# HiFE: Hierarchical Feature Ensemble Framework for Few-Shot Hypotheses Adaptation

**Yongfeng Zhong** *                                          *csyzhong@comp.polyu.edu.hk*
*Department of Data Science and Artificial Intelligence*
*The Hong Kong Polytechnic University*

**Haoang Chi** *                                             *haoangchi618@gmail.com*
*Intelligent Game and Decision Lab, Defense Innovation Institute*
*National University of Defense Technology*

**Feng Liu**                                                 *fengliu.ml@gmail.com*
*Computing and Information Systems*
*University of Melbourne*

**Xiaoming Wu** †                                            *xiao-ming.wu@polyu.edu.hk*
*Department of Data Science and Artificial Intelligence*
*The Hong Kong Polytechnic University*

**Bo Han**                                                   *bhanml@comp.hkbu.edu.hk*
*Department of Computer Science*
*Hong Kong Baptist University*

**Reviewed on OpenReview:** *https://openreview.net/forum?id=B6RS6DNOGt*

## Abstract

Transferring knowledge from a source domain to a target domain in the absence of source data constitutes a formidable obstacle within the field of source-free domain adaptation, often termed hypothesis adaptation. Conventional methodologies have depended on a robustly trained (strong) source hypothesis to encapsulate the knowledge pertinent to the source domain. However, this strong hypothesis is prone to overfitting the source domain, resulting in diminished generalization performance when applied to the target domain. To mitigate this issue, we advocate for the augmentation of transferable source knowledge via the integration of multiple (weak) source models that are underfitting. Furthermore, we propose a novel architectural framework, designated as the Hierarchical Feature Ensemble (HiFE) framework for Few-Shot Hypotheses Adaptation, which amalgamates features from both the strong and intentionally underfit source models. Empirical evidence from our experiments indicates that these weaker models, while not optimal within the source domain context, contribute to an enhanced generalization capacity of the resultant model for the target domain. Moreover, the HiFE framework we introduce demonstrates superior performance, surpassing other leading baselines across a spectrum of few-shot hypothesis adaptation scenarios.

---

*Equal contribution.
†Corresponding author.

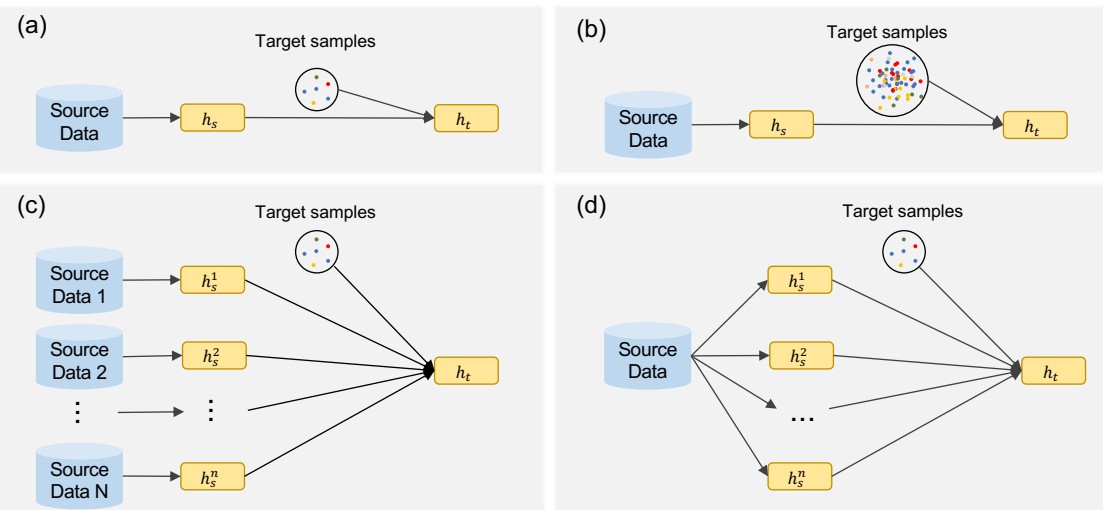

Figure 1: Different approaches to improve the performance of Few-shot Hypothesis Adaptation (FHA). (a) Conventional FHA. (b) Enhancing target performance by increasing the number of target samples. (c) Typical ensemble approaches to improve target performance by increasing the number of source domains. (d) Improving target performance under resource constraints with only one source domain and a limited number of target samples by generating multiple source hypotheses from the given source domain.

## 1 Introduction

Domain adaptation (DA) (Ben-David et al., 2010) refers to the study of leveraging labeled data in a *source domain* (SD) to obtain a predicted model for a given *target domain* (TD) where labels are insufficient or unavailable. Conventional DA methods (Ahmed et al., 2021; Jiang et al., 2021; Kang et al., 2019; Sukhija et al., 2016; Wang et al., 2019) pose a potential risk of exposing private information caused by accessing the source data. To mitigate this concern, recent studies have introduced source-free DA, also referred to as *hypothesis adaptation* (HA) (Liang et al., 2020; Li et al., 2020; Yang et al., 2021; Yi et al., 2023), which leverages a source model to encode the knowledge from the SD rather than the source data. Recently, *few-shot HA* (FHA) (Chi et al., 2021; Yazdanpanah & Moradi, 2022), which operates effectively in scenarios with limited labeled data from the TD, has emerged as an appealing approach to address data scarcity (Figure 1 (a)). To improve the FHA performance, recent approaches have attempted to increase the size of target samples via data labeling or generation (Figure 1 (b)) or gather multiple source domains to generate more source hypotheses (Figure 1 (c) (Ahmed et al., 2021; Shu et al., 2022; Li et al., 2024)).

However, the abovementioned approaches require additional effort to label the target data or to collect hypotheses from different source domains. Moreover, these approaches rely on best-performing strong source models from the SD, which may overfit the SD and subsequently perform worse on the TD after adaptation. This overfitting issue has been demonstrated by a pilot experiment in which models with varying accuracies from the digit dataset SVHN were adapted to the target task Mnist using a straightforward fine-tuning approach (details are provided in Appendix A). In this experiment, under-fitted weak source models may exhibit superior performance compared to the strong model after adaptation (Figure 4). This indicates that **some weak hypotheses, although suboptimal for the SD, may contain underlying knowledge that could be beneficial for the TD**. Inspired by this experiment, we propose addressing the FHA problem by generating multiple source hypotheses with varying source accuracies from a single SD (Figure 1 (d)). Notably, the routine training of source models naturally produces a series of weak intermediate models that are often overlooked and discarded. These models are easily obtainable, as reaching out to the source provider to acquire additional weak models incurs minimal additional cost.

**A new problem: FHAW.** Thus, we propose to study a new problem called *few-shot hypotheses adaptation with weak models* (FHAW). Unlike previous single-source FHA approaches (Chi et al., 2021; Liang et al., 2020)

that depend on a single strong hypothesis, FHAW aims to leverage multiple source hypotheses with varying degrees of accuracy from a single SD to enhance the diversity of source models. While some researchers have advocated for utilizing multi-source hypotheses for HA (Ahmed et al., 2021; Shu et al., 2022; Li et al., 2024), these ensemble methods typically rely on multiple models originating from different source domains, each contributing its best-performing hypothesis to the ensemble (Figure 1 (b)). This approach becomes ineffective when dealing with weak hypotheses that have low source accuracies. FHAW addresses a more challenging scenario where access to multiple source domains is not feasible, offering a new perspective on tackling FHA challenges.

**A viable approach to solve FHAW**. To mitigate the potential negative transfer arising from weak source models and extract valuable source knowledge for the target task, we have developed a new and efficient framework called the Hierarchical Feature Ensemble framework (HiFE) to address FHAW, as illustrated in Figure 2. The HiFE method employs hierarchical ensemble techniques to enhance the representativeness of intermediate features. It utilizes weighted residual units (WRU) to aggregate features induced by the source hypotheses, as illustrated in Figure 2 (b). WRU merges similar features with skip connections to reduce the risk of forgetting the source knowledge, alleviating the overfitting problem when fine-tuning with few-shot samples. Furthermore, we incorporate feature decorrelation learning (DeCL) by integrating a correlation penalty term into the standard classification loss, thereby enhancing the diversity of intermediate features (refer to Figure 2 (c)). Comprehensive results indicate that HiFE delivers state-of-the-art (SOTA) performance across various domain adaptation tasks.

**Main contributions**. Our contributions are three-fold:

- To the best of our knowledge, this is the initial investigation into the FHAW problem. FHAW holds practical relevance in numerous private data-based scenarios, as source providers are inclined to offer "redundant" weak models instead of disclosing sensitive datasets. Our work introduces a fresh perspective to encode the source knowledge in the absence of the source data.
- We propose a new framework to aggregate all source hypotheses at the feature level to address the FHAW problem. We are the first to apply a hierarchical ensemble at the feature level in hypotheses adaptation. We effectively alleviate the over-fitting problem by the design of WRU and improve the generalization of the final hypothesis by incorporating feature DeCL loss under the few-shot setting.
- The comprehensive evaluation of the proposed HiFE methodology, conducted over an array of benchmark datasets—including Mnist, SVHN, USPS, CIFAR-10, STL-10, Amazon, DSLR, Webcam, and VisDA-C-has established that our approach achieves performance on par with or exceeding current SOTA methods in various FHA tasks. Notably, as detailed in Table 2, the HiFE method surpasses the SOTA by an average accuracy of 4.3% in the digit dataset task USPS $\rightarrow$ Mnist. Similarly, in the task of adapting DSLR to Webcam datasets, as shown in Table 3, HiFE outperforms the SOTA by 3.6% in accuracy.

## 2 Related Work

This section presents a brief overview of the literature about traditional domain adaptation, hypothesis adaptation, multi-hypotheses adaptation, and ensemble methods for hypothesis adaptation.

**Domain adaptation (DA)**. Traditional DA is a subfield of machine learning that focuses on learning a hypothesis for a TD when labeled data is insufficient or unavailable by leveraging labeled data from an SD. Numerous DA methods have been proposed for various tasks such as object classification (Liang et al., 2018), object detection (Hsu et al., 2020), and semantic segmentation (Zou et al., 2018). Existing approaches for DA can mainly be categorized into two classes: feature-based DA and instance-based DA. The former aims to learn a domain-invariant representation by minimizing the domain discrepancy in a shared space (Kang et al., 2019; Long et al., 2017). For example, Gradually Vanishing Bridge (Cui et al., 2020) uses bi-directional generation to learn domain-invariant representations. The latter minimizes the discrepancy by re-weighting the source samples for better training. Despite the success achieved by these methods, they require access to source data during the learning process, which incurs significant costs in terms of data transfer and storage as well as risks related to personal information leakage.

**Hypothesis adaptation (HA)**. Researchers have started exploring source-free domain adaptation (SFDA), namely HA, to mitigate the issues arising from accessing source data. Early works addressed the problem by fine-tuning the source hypothesis on the target data (Girshick et al., 2014). However, recent studies have delved into unsupervised DA to investigate the limitations of this straightforward strategy (Ding et al., 2022; Liang et al., 2020; Yang et al., 2022; Yi et al., 2023). Among these methods, SHOT (Liang et al., 2020) proposes a representation learning framework to update the feature extractor through information maximization and self-supervised pseudo-labeling loss. In this framework, pseudo-labels of the target data are refined using the nearest centroids. Similarly, (Yi et al., 2023) views SFDA as the problem of learning with label noise and suggests exploiting the early-time training phenomenon to tackle the issue of pseudo-labels. Notably, these methods rely on large amount of unlabeled data from the TD to purify the pseudo-labels. On the other hand, TOHAN (Chi et al., 2021) is the first study to explore the HA under a few-shot setting. It proposes generating an intermediate domain that is compatible with the TD to facilitate transfer learning. Many previous works rely on a strong source hypothesis for adaptation, which may not always be the most suitable one for adapting to a specific TD.

**Multi-hypotheses adaptation (MHA)**. MHA extends the HA paradigm by integrating knowledge from source hypotheses from multiple domains. To tackle this, model selection methods (Nguyen et al., 2020; You et al., 2021) have been developed to estimate the transferability of each pre-trained hypothesis. However, the single selected hypothesis may not be able to carry the rich knowledge encapsulated in all of the source hypotheses. Thus, some researchers have turned to parameter ensemble methods (Ahmed et al., 2021; Rusu et al., 2016; Shu et al., 2022; Li et al., 2024). Yet, these approaches often require significant amounts of unlabeled target data to be effective, making them less tenable in the FHA setting. Moreover, these approaches operate under the assumption that each source hypothesis is a strong one from the corresponding SD, rendering them ineffective when presented with weak hypotheses. Besides, these approaches require accessing multiple source domains related to the TD, which is often not feasible. Our research focuses on a more practical and challenging scenario: only one SD is available.

**Ensemble methods for HA**. Ensemble methods are prominent research in machine learning (Dietterich, 2000; Dong et al., 2020; Sagi & Rokach, 2018; Eilers et al., 2022). These methods have demonstrated that combining multiple hypotheses is advantageous over a single hypothesis in classification and regression problems. However, traditional ensemble methods rely on weighted voting for the final decision and lack the capability of representation learning (Cao et al., 2012). Thus, some researchers have proposed feature-level ensembling. Studies have demonstrated the effectiveness of hierarchical feature representation in improving classification accuracy (Cai et al., 2018; Su et al., 2009). In the area of FHA, research on developing hierarchical feature-level ensemble methods to derive a comprehensive knowledge representation of all source hypotheses has been limited.

## 3 Few-Shot Hypotheses Adaptation with Weak Models

### 3.1 Problem Definition

We address the problem of few-shot hypotheses adaptation with weak models, where several pre-trained source hypotheses, including one strong and some weak hypotheses, are given. Let $\mathcal{X} \subset \mathbb{R}^d$ be an input space and $\mathcal{Y} := \{1, \ldots, C\}$ be the label space, where $C$ is the number of classes. To formalize the problem clearly, some definitions are presented as follows.

**Definition 1.** (Expected and empirical risk). Given a data distribution $P$ over $\mathcal{X} \times \mathcal{Y}$, let $\mathcal{H} = \{h : \mathcal{X} \to \mathcal{Y}\}$ be the hypothesis space and $h \in \mathcal{H}$ with the parameter $\theta \in \Theta$, then the expected and empirical risks are defined as

$$L(\theta) = \mathbb{E}_{(x,y) \sim P}[\ell(\theta, x, y)],$$

$$\hat{L}(\theta, D) = \frac{1}{n} \sum_{i=1}^{n} (\ell(\theta, x_i, y_i)),$$

where $\ell$ is a proper loss function and $D = \{(x_i, y_i)\}_{i=1}^{n} \sim P^n$ denotes the *i.i.d.* $n$ observations.

**Definition 2.** (Strong Hypothesis). Given a set of hypotheses $\hat{\mathcal{H}} = \{h^m\}_{m=1}^M$ from domain $S$ and a validation set $D_{\text{val}}$, where $m$ is the hypothesis ID and $M$ is the number of hypothesis, a hypothesis $h_s \in \hat{\mathcal{H}}$ is called a strong hypothesis if $\forall h \in \hat{\mathcal{H}}$, $\hat{L}(\theta_{h_s}, D_{\text{val}}) \leq \hat{L}(\theta_h, D_{\text{val}})$.

**Definition 3.** (Weak Hypothesis). Given a set of hypotheses $\hat{\mathcal{H}} = \{h^m\}_{m=1}^M$ from domain $S$ and a validation set $D_{\text{val}}$, where $m$ is the hypothesis ID and $M$ is the number of hypothesis, a hypothesis $h_w \in \hat{\mathcal{H}}$ is called a weak hypothesis if $\exists h \in \hat{\mathcal{H}}$, $\hat{L}(\theta_h, D_{\text{val}}) < \hat{L}(\theta_{h_w}, D_{\text{val}})$.

**Problem 1.** (Few-Shot Hypotheses Adaptation with Weak Models (**FHAW**)). Given a set of hypotheses $\hat{\mathcal{H}}$ with a strong hypothesis $h_s$ and $M$ weak hypotheses $\{h_w^m\}_{m=1}^M$ trained on the SD $P_S(X, Y)$, $n_t$ target labeled data $D_t = \{(x_t^i, y_t^i)\}_{i=1}^{n_t}$ that *i.i.d.* drawn from $P_T(X, Y)$ with $n_t \ll n_s$ and $P_S(X, Y) \neq P_T(X, Y)$, FHAW is to learn a target hypothesis $h_t : \mathcal{X} \to \mathcal{Y}$ with $h_s$, $\{h_w^m\}_{m=1}^M$ and $D_t$ to minimize the expected risk on the TD.

**Comparison with FHA**. FHA involves the use of a strong hypothesis derived from the SD. However, such a hypothesis is prone to over-fitting on the SD, and their generalizability towards the TD can be limited. To address this challenge, we introduce FHAW by leveraging multiple weak hypotheses to facilitate more effective adaptation. These weak hypotheses can be easily obtained by saving model snapshots during the training of the strong source model with minimal additional cost.

## 3.2 Addressing FHAW in Principle

We will present a theoretical view based on the PAC-Bayesian framework (Germain et al., 2009; McAllester, 1999; Masegosa, 2020) to demonstrate why we propose to incorporate multiple weak hypotheses for FHA and why our HiFE framework works. In the PAC-Bayesian framework, each hypothesis $h_\theta$ has prior knowledge of the hypothesis space $\Theta$, and this prior distribution $\pi$ is updated to a posterior distribution $\rho$ after feeding samples $D$ to $h_\theta$. In FHAW, multiple models $\{h_{\theta_i}\}_{i=1}^M$ are given with $\theta_i \in \Theta_i$, $\theta = \{\theta_i\}_{i=1}^M$ and $\rho(\theta) = \prod_{i=1}^M \rho_i(\theta_i)$. For a given sample $(x, y)$, we apply the cross-entropy loss $\ell(\theta, x, y) = -\log p(y|x, \theta)$. A bound theorem proposed by Deng et al.for the model ensemble is restated below, and some other previous related theorems are shown in Appendix C.

**Theorem 1.** *(Model ensemble error bound (Deng et al., 2023)). Given a data distribution $P$ over $\mathcal{X} \times \mathcal{Y}$, a set of model parameters $\{\Theta_i\}_{i=1}^M$ with associated prior $\{\pi_i\}_{i=1}^M$, where $\pi_i$ is defined over $\Theta_i$ with $\pi_i(\theta_i) \sim \mathcal{N}(0, \sigma^2 I)$, a $\delta \in (0, 1]$, a real number $c > 0$, and $\rho_i(\theta_i)$ is a Dirac-delta distribution centered around $\theta_i'$ with $\rho_i(\theta_i) = \delta_{\theta_i'}(\theta_i)$, then we have that the $\mathbb{E}_{\rho(\theta)}(L(\theta))$ is upper bounded by*

$$\frac{1}{M} \sum_{i=1}^M \left( \hat{L}(\theta_i', D) + \frac{1}{2cn\sigma^2} \| \theta_i \|^2 + \frac{d_i}{2cn} \log(2\pi\sigma^2) \right) - \hat{\mathbb{V}}(\rho(\theta), D) + \frac{\epsilon}{cnL},$$

*where $\hat{\mathbb{V}}(\rho(\theta), D)$ is the empirical version of a variance term $\mathbb{V}(\rho(\theta))$, which is defined as*

$$\mathbb{E}_{\rho(\theta)} \mathbb{E}_{(x,y) \sim P} \left[ \frac{1}{2M \max_\theta p(y|x, \theta)^2} \sum_{i=1}^M \left( p(y|x, \theta_i) - \frac{1}{M} \sum_{k=1}^M p(y|x, \theta_k) \right)^2 \right],$$

*and $\epsilon$ is defined as*

$$\log \frac{\mathbb{E}_{\pi(\theta)} \mathbb{E}_{D \sim P^n} \left[ e^{cn \left( \sum_{i=1}^M \left( L(\theta_i) - \hat{L}(\theta_i, D) \right) - M \left( \mathbb{V}(\theta) - \hat{\mathbb{V}}(\theta, D) \right) \right)} \right]}{\delta}.$$

In Theorem 1, the variance term $\hat{\mathbb{V}}(\rho(\theta), D)$ measures the diversity of all models (Masegosa, 2020). If there exists an input sample $x$ such that $h_{\theta_i}(x) \neq h_{\theta_j}(x)$, then we have $\hat{\mathbb{V}}(\rho(\theta), D) > 0$. Therefore, in the setting of FHAW, adding weak hypotheses increases the diversity of the source models and provides opportunities to decrease this error bound. Minimizing the first term of the error bound in Theorem 1 is equivalent to finding $\theta = \{\theta_i\}_{i=1}^M$ by $\min_\theta \sum_{i=1}^M \left( \hat{L}(\theta_i, D) + \lambda_1 \| \theta_i \|^2 + \lambda_2 d_i \right) / M$, where $d_i$ is the dimension of $\theta_i$ and $\lambda_1$, $\lambda_2 > 0$ are hyper-parameters. Based on this formula, we propose a hierarchical feature ensemble module to reduce the dimensionality of features.

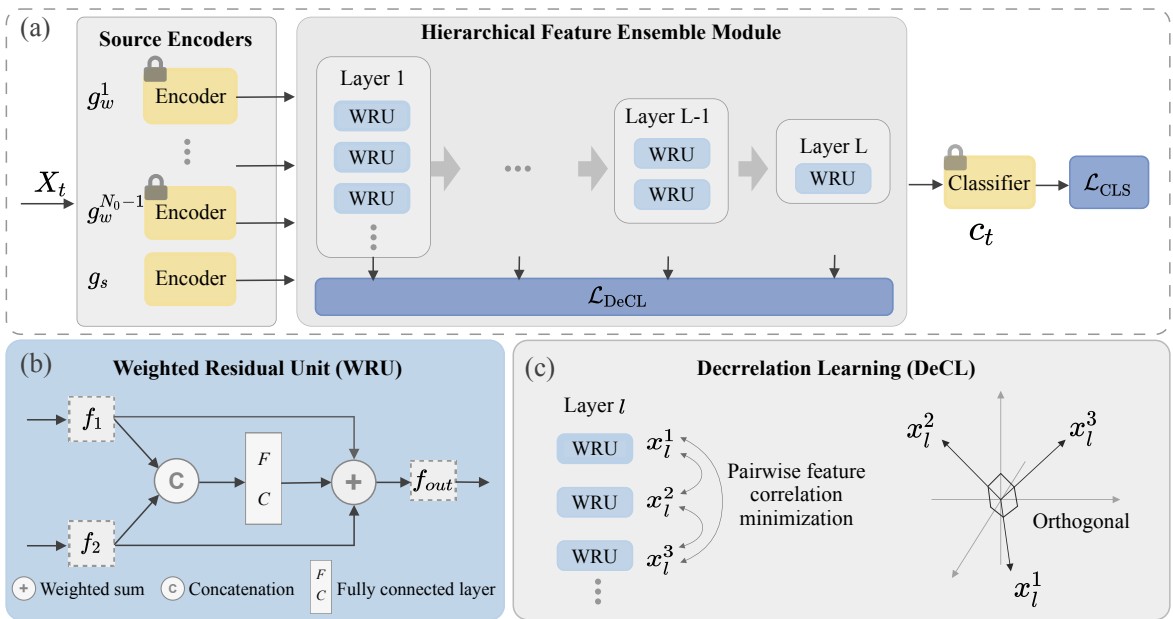

Figure 2: The architecture of HiFE framework. Each source hypothesis consists of a feature encoder and a classifier. We train a model with a hierarchical feature ensemble module to merge features from all source encoders. In this module, features are grouped according to the cosine similarity, and WRU merges the grouped features to generate new features for the next layer. Each WRU leverages skip connections to avoid over-fitting. Besides, we apply the decorrelation learning (DeCL) strategy by adding a correction penalty term to the loss function to encourage feature diversity. The target classifier $c_t$, initializing with the average of all the source classifiers, is fixed during the training. Only the parameters of the strong encoder $g_s$ and the hierarchical feature ensemble module are updated.

## 4 Few-shot Hypotheses Adaptation via Hierarchical Feature Ensemble

To aggregate knowledge from both strong and weak source hypotheses, we introduce the HiFE framework, depicted in Figure 2. HiFE hierarchically merges features induced by all the source hypotheses. We assume each source hypothesis has been embedded with its specific discriminative knowledge about the SD. Hence, during the aggregation, we use a feature de-correction learning module, making the features as mutually independent as possible at each level to increase the representative power of the intermediate features. We describe the design insights of HiFE in Section 4.1 and Section 4.2.

### 4.1 Hierarchical Feature Ensemble

Ensemble learning is widely recognized as an effective approach for combining multiple learning methods and improving overall performance (Beven & Binley, 1992; Kuczera & Parent, 1998). While ensemble methods have been utilized for HA in past research (Ahmed et al., 2021), ensemble learning at the feature representation level has received relatively less attention. However, prior research has shown that hierarchical feature representations can significantly enhance classification accuracy. We propose a hierarchical feature ensemble-based approach for FHAW to leverage such benefits. Specifically, our method involves merging source features that contain knowledge of the SD using a *hierarchical feature ensemble module* before feeding them to the final classifier.

To simplify the feature extraction process with the source hypotheses, we follow (Motiian et al., 2017) and (Ahmed et al., 2021) to decompose each hypothesis $h$ into two modules: a feature encoder $g : \mathcal{X} \to \mathbb{R}^d$ and a classifier $c : \mathbb{R}^d \to \mathbb{R}^C$, where $d$ denotes the dimension of the output feature. Thus, we have $h(x) = c(g(x))$,

i.e., $h = c \circ g$. In our problem setting, source hypotheses are decomposed to $c_s \circ g_s$ and $\{c_w^m \circ g_w^m\}_{m=1}^M$. Features induced by $g_s$, $\{g_w^m\}_{m=1}^M$ are fed into the feature ensemble module. Let $x_l^i$ denote the $i$-th feature at layer $l$ ($l \in \{0, 1, ..., L\}$) and $\{x_0^i\}_{i=1}^{N_0}$ be the $N_0$ input features. The hierarchical feature ensemble module aims to aggregate all these features into one single feature $x_L^1$ through a hierarchical method so that $x_L^1$ contains as much source knowledge as possible. To this end, we must tackle two questions: 1) which features to merge and 2) how to merge the chosen features.

1) *Which features to merge?* According to the Gestalt principles of psychology (Koffka, 2013), humans tend to group similar information during cognitive processing. Taking inspiration from this, we utilize feature similarity as a metric to group similar input features together from the previous layer for the purpose of merging. Given a set of $N_l$ features, we first create a similarity matrix $S \in \mathbb{R}^{N_l \times N_l}$, where $S_{i,j} = \cos(x_l^i, x_l^j)$ is the cosine similarity of features $x_l^i$ and $x_l^j$. Next, we repeatedly choose two or more features with the highest similarity and merge them into a new feature for the next layer. The merging is done only between features from the previous layer. Such a hierarchical merge process repeats layer by layer until only one output feature is left.

2) *How to merge features?* In the FHA setting, the small sample size problem limits the feasibility of maintaining adequate validation sets to assess performance before testing unknown samples. Without such validation sets, optimizing the model could cause over-fitting to the limited target data, leading to a local optimum and performance degradation (Goodfellow et al., 2014; Kirkpatrick et al., 2017). To address this issue, we propose the *Weighted Residual Unit* (WRU), adding the "shortcut connections" of the input features to the block output after feature merging (see Figure 2 (b)). The shortcut connections allow the upper layer's features to be directly sent to the next layer, maintaining the source knowledge during adaptation and alleviating the over-fitting problem. Within each WRU, the input features $\{f_i\}_{i=1}^K$ are concatenated and fed into a fully connected (FC) layer, where $K$ is the number of input features. The output of the FC layer is denoted as $\text{FC}(\{f_i\}_{i=1}^K, W_{\text{FC}})$ with $W_{\text{FC}}$ be the learnable parameters of this layer. Unlike the shortcut connections that perform identity mapping in ResNet (He et al., 2016), we perform a weighted element-wise addition of $\{f_i\}_{i=1}^K$ and $\text{FC}(\{f_i\}_{i=1}^K, W_{\text{FC}})$ to balance the influence of different input features induced by the source hypotheses. The function of WRU can be formalized as follows.

$$\text{WRU}(\{f_i\}_{i=1}^K) = \alpha_0 \cdot \text{FC}(\{f_i\}_{i=1}^K, W_{\text{FC}}) + \sum_{i=1}^K \alpha_i \cdot f_i, \tag{1}$$

where $\{\alpha_i\}_{i=0}^K$ are the learnable weights. We add the batch normalization and ReLu layers after the weighted sum for better performance. If the dimension of $f_i$ is not equal to that of the output of FC, we can make a linear projection of $f_i$ by extending $\alpha_i$ to a square matrix $W_i$ to match the dimension. The application of WRU allows us to preserve some source knowledge and learn new information from the target samples simultaneously.

In our approach, we follow the aforementioned principle to determine which features to merge and then use the WRU to merge the features based on the first batch of input data. Once we pass the first batch of input data, the merging network is built. This merging network then remains fixed, and all subsequent samples share the same merging network. Such a setting ensures stable fine-tuning on target samples.

## 4.2 Decorrelation Learning

It has been commonly agreed that diversity is a success factor of ensemble algorithms. Different opinions from multiple classifiers are expected to reduce the generalization error. Traditional decorrelation learning (DeCL) methods encourage diversity explicitly by adding a correlation penalty term to the final error function (Liu & Yao, 1999; Shi et al., 2018; Wang et al., 2010). When it comes to feature ensemble, learning the features with good discriminative power is also essential for various high-level vision tasks (Wen et al., 2016; Cheng et al., 2018). To promote the learning of features widely distributed across the feature space and embed various forms of source knowledge, we apply DeCL in the feature space to encourage independence between features in each layer. In this regard, we introduce a cosine similarity penalty to decrease feature correlation and encourage feature diversity (see Figure 2 (c)). Specifically, we calculate the pairwise square values of

cosine similarities for all features in the same layer and sum them up from all layers. The corresponding feature DeCL loss is defined as

$$\mathcal{L}_{\text{DeCL}} = \sum_{l=1}^{L} \sum_{i=1}^{N_l-1} \sum_{j=i+1}^{N_l} \cos(x_l^i, x_l^j)^2, \tag{2}$$

where $N_l$ is the number of features at layer $l$ and $\cos(x_l^i, x_l^j) = (x_l^i \cdot x_l^j)/(\|x_l^i\| \cdot \|x_l^j\|)$. Furthermore, to enable the adaptation of the ensemble network to the TD, we incorporate the knowledge of TD by fitting the network to the labeled target data. To accomplish this, we adopt the standard cross-entropy loss, which is defined as follows,

$$\mathcal{L}_{\text{CLS}} = \mathbb{E}_{(x_t, y_t) \sim P_T}[\text{CE}(c_t(A(x_t)), y_t)], \tag{3}$$

where $\text{CE}(\cdot)$ denotes the cross-entropy loss and $A(x_t)$ refers to the output of the feature ensemble module when fed $x_t$ to the source encoders. To summarize, we train the ensemble network using joint supervision that combines the target supervised loss (Equation (3)) and a feature DeCL penalty term (Equation (2)) with a hyper-parameter $\beta$ to trade off the two aspects (Equation (4)). The target supervised loss guides the network in learning the knowledge from the target samples, while the feature DeCL loss promotes mutual independence amongst features in each layer, thereby increasing the feature diversity and preserving the distinct discriminative knowledge of each source hypothesis.

$$\mathcal{L}(\beta) = (1 - \beta) \cdot \mathcal{L}_{\text{CLS}} + \beta \cdot \mathcal{L}_{\text{DeCL}}. \tag{4}$$

## 5 Experiments

### 5.1 Experimental Setup

**Datasets**. We conduct experiments on various standard DA benchmarks to evaluate our approach[1]

*Digits.* We choose three-digit datasets, i.e., Mnist (**M**), USPS (**U**), and SVHN (**S**) for our experiments. Following (Motiian et al., 2017; Chi et al., 2021), we experiment with different numbers of target samples from 1 to 7 per class.

*Office.* We use three domains of the office datasets (Saenko et al., 2010): Amazon (**A**), DSLR (**D**), and Webcam (**W**). Each domain contains 31 object classes in the office environment. We conduct several experiments with different numbers of target samples per class ranging from 1 to 5.

*Image classification.* We use two image classification benchmarks CIFAR-10 (**CF**) (Krizhevsky, 2009) and STL-10 (**ST**) (Coates et al., 2011). Each benchmark consists of 10 classes of objects, and nine classes are overlapped. We remove the non-overlapped classes ("monkey" and "frog") and reduce the tasks to a 9-class classification problem following the procedure in (Shu et al., 2018). As the two domains are more complex than digits, we increase the number of target samples to 15 and 20 for each class.

*VisDA-C.* VisDA-C (Peng et al., 2017) is a demanding large-scale benchmark designed primarily for the 12-class synthesis-to-real object recognition task. The source domain comprises 152,000 synthetic images created by rendering 3D models from different angles and with different lighting conditions, whereas the target domain includes 72,000 real object images sourced from Microsoft COCO. We randomly choose 10% of the target data set (7200 images) as the testing set. Since the domain gap between the synthesis and real-object images is large, we experiment on a larger number (10, 30, and 50) of the target samples.

**Baseline methods**. In the context of the novel FHAW problem setting, we establish our baseline comparisons by adapting and refining several established approaches in the field. We conducted a comprehensive evaluation against four existing methods for HA and their respective variations. Initially, SHOT (Liang et al., 2020), a hypothesis transfer learning framework tailored for unsupervised HA, served as a foundation. In our

---

[1]The full code is available at `https://github.com/yfZhong/HIFE.git`.

study, we preserved its model adaptation module, tweaking it to leverage labeled target data to support supervised HA, aligning it seamlessly with our experimental setup. The performance outcomes from employing solely the strong hypothesis with SHOT are denoted as *SHOT-strong*. The subsequent contender, TOHAN (Chi et al., 2021), specifically tackles the FHA challenge. Both SHOT and TOHAN are engineered to adapt a singular hypothesis to the TD independently. To evaluate against our multi-hypotheses adaptation approach, we extended these methods to *SHOT-ens* and *TOHAN-ens* by employing a straightforward ensemble technique, following the methodology outlined in (Ahmed et al., 2021). Furthermore, we included the models DECISION (Ahmed et al., 2021) and Bi-ATEN (Li et al., 2024), specifically designed for multi-source-free unsupervised HA, to contrast with our single-source multi-model HA strategy.

**Network architecture**. For digit recognition tasks, we employ the same architectures utilized in SHOT (Liang et al., 2020), namely using the LeNet-5 (LeCun et al., 1998) for Mnist, USPS, and a modified version of LeNet for the slightly more complex SVHN dataset. For the image classification tasks, we adopt ResNet-18, ResNet-50, and ResNet-101 ((He et al., 2016)) as the backbones for the CIFAR-10/STL-10, office, and VisDA-C datasets, respectively.

**Source hypotheses preparation**. We train a single optimal hypothesis as a strong source hypothesis for each SD and save seven intermediate snapshots as weak source hypotheses with varying accuracy levels. To acquire the hypotheses with different source accuracies, we first set an accuracy range $[\text{acc}_{min}, \text{acc}_s]$, where $\text{acc}_{min}$ is a preset value around at 40-60% and $\text{acc}_s$ is an estimation of the accuracy of the strong hypothesis. Then, we split this range into several uniform intervals and save one model snapshot at each interval to get weak hypotheses for each SD. The source data can be discarded after getting all the required source hypotheses. Section 5.1 shows the source models generated with the source dataset Mnist and their corresponding accuracy ranges. We generate 12 source models $\{h_i|_{i=1}^{12}\}$. According to our definition in Section 3, the first 11 models $\{h_i|_{i=1}^{11}\}$ are weak source hypotheses, while the last one $h_{12}$ is the strong hypothesis. Among these models, we used $\{h_i|_{i=5}^{12}\}$ (8 models) in the experiments shown in Table 2 and Table 8, while all models are prepared for ablation studies. The process of training the target ensemble hypothesis with HiFE is detailed in Appendix B.

| MI | AC | MI | AC | MI | AC | MI | AC |
|---|---|---|---|---|---|---|---|
| $h_1$ | [40, 45) | $h_4$ | [55, 60) | $h_7$ | [70, 75) | $h_{10}$ | [85, 90) |
| $h_2$ | [45, 50) | $h_5$ | [60, 65) | $h_8$ | [75, 80) | $h_{11}$ | [90, 95) |
| $h_3$ | [50, 55) | $h_6$ | [65, 70) | $h_9$ | [80, 85) | $h_{12}$ | [95, 100) |

Table 1: An example demonstrating the generation of source hypotheses using the Mnist dataset. We first set an accuracy range of [40%, 100%]. This range is then divided into 12 uniform intervals. During training on the Mnist training set, we save one model snapshot at each interval to obtain the corresponding hypotheses.

## 5.2 Result Analysis

**Results of digit classification tasks.** We evaluate the effectiveness of our approach on six closed-set adaptation tasks for digit classification. These tasks are by pairwise combinations of the three domains $S$, $M$, and $U$. We report the results of three tasks in Table 2 (more results can be found in Appendix D). Firstly, as shown in Table 2, there exist some weak hypotheses that can perform better than the strong hypothesis after adaptation (see the comparison of SHOT-best and SHOT-strong), supporting our motivation of adopting the weak hypotheses. Moreover, incorporating the weak hypotheses allows our proposed HiFE to outperform SHOT-strong. For instance, compared with the average accuracy of SHOT-strong (77.3%), HiFE leads to higher average accuracy (88.7%) in the task $S \to M$. Additionally, despite some weak hypotheses with bad adaptation performance (see SHOT-worst), HiFE can largely avoid the severe negative transfer and achieve the best performance than previous ensemble approaches. For example, HiFE outperforms the SOTA (TOHAN-ens) by 4.3% in the average accuracy of $U \to M$ task. Notably, we can observe that our approach not only maintains competitive accuracy but also decreases the standard deviation (std) of accuracy across different target samples. This reduction in std is paramount as it indicates a more consistent and reliable performance of our HiFE.

**Results of office object classification tasks**. We show the results of three closet-set adaptation tasks

| Tasks | Hypothesis Number | Method | Number of Target Data per Class | | | | | | | Avg |
|---|---|---|---|---|---|---|---|---|---|---|
| | | | 1 | 2 | 3 | 4 | 5 | 6 | 7 | |
| $U \to M$ | Single | SHOT-worst | 42.1±1.2 | 44.3±0.8 | 49.9±1.0 | 48.4±1.6 | 50.7±0.6 | 50.9±1.1 | 50.9±0.8 | 48.2 |
| | | SHOT-best | 92.1±1.5 | 93.4±1.2 | 93.7±0.9 | 93.6±1.0 | 93.7±1.5 | 93.5±0.8 | 94.0±0.6 | 93.4 |
| | | SHOT-strong | 89.8±1.1 | 90.3±1.3 | 92.0±1.5 | 91.3±1.6 | 92.0±0.7 | 92.0±0.8 | 91.9±0.5 | 91.3 |
| | Multiple | SHOT-ens | 86.8±1.6 | 88.5±1.8 | 90.0±2.1 | 89.5±2.3 | 90.5±1.9 | 90.6±1.0 | 90.8±1.3 | 89.5 |
| | | TOHAN-ens | 87.3±1.8 | 89.7±1.6 | 90.1±1.6 | 90.5±1.4 | 91.2±1.5 | 92.5±0.9 | 93.5±0.7 | 90.7 |
| | | DECISION | 88.7±2.3 | 88.8±1.8 | 89.6±2.4 | 89.8±2.1 | 90.3±1.7 | 90.2±1.3 | 90.5±1.1 | 89.7 |
| | | Bi-ATEN | 89.5±1.1 | 90.9±1.2 | 91.5±0.8 | 91.1±0.4 | 92.1±1.2 | 92.5±2.3 | 90.5±2.6 | 91.2 |
| | | HiFE (ours) | **92.7±0.8** | **94.9±0.2** | **95.0±0.4** | **95.2±0.6** | **95.4±0.5** | **95.4±0.7** | **96.1±0.3** | **95.0** |
| $S \to M$ | Single | SHOT-worst | 40.9±1.0 | 45.1±1.2 | 50.9±1.1 | 51.6±0.9 | 51.7±1.1 | 51.8±0.8 | 51.9±0.8 | 49.1 |
| | | SHOT-best | 74.8±1.4 | 75.1±1.2 | 79.8±1.3 | 79.1±0.9 | 80.6±1.1 | 79.8±0.5 | 79.1±0.6 | 78.3 |
| | | SHOT-strong | 74.5±2.0 | 73.5±1.1 | 78.7±1.8 | 78.2±1.5 | 78.8±1.3 | 78.6±0.9 | 78.7±0.8 | 77.3 |
| | Multiple | SHOT-ens | 75.6±2.2 | 74.9±1.2 | 81.2±2.6 | 81.5±1.4 | 82.0±1.3 | 81.6±1.0 | 81.7±1.5 | 79.8 |
| | | TOHAN-ens | 79.0±1.9 | **85.9±2.1** | 87.5±1.6 | 89.5±1.1 | 90.1±1.4 | 90.6±1.2 | 91.1±0.9 | 87.7 |
| | | DECISION | 71.9±1.3 | 72.1±2.1 | 72.5±2.0 | 73.4±1.5 | 75.0±1.2 | 76.7±1.5 | 79.2±1.0 | 74.4 |
| | | Bi-ATEN | 75.1±1.7 | 77.2±1.1 | 77.1±2.3 | 79.7±2.5 | 80.1±2.9 | 82.5±1.9 | 83.1±1.6 | 79.3 |
| | | HiFE (ours) | **79.2±2.1** | 85.7±2.0 | **88.1±1.0** | **90.3±0.9** | **92.2±0.7** | **92.5±0.9** | **92.8±1.0** | **88.7** |
| $U \to S$ | Single | SHOT-worst | 15.8±2.1 | 14.8±1.9 | 14.7±0.9 | 14.3±1.5 | 14.3±1.7 | 14.0±0.9 | 14.4±0.5 | 14.6 |
| | | SHOT-best | 32.6±1.1 | 32.4±1.6 | 34.5±1.2 | 37.3±2.0 | 38.4±0.8 | 40.6±0.6 | 40.5±0.7 | 36.6 |
| | | SHOT-strong | 32.6±1.1 | 32.3±1.7 | 34.3±1.6 | 37.0±1.3 | 38.2±0.9 | 40.2±0.8 | 40.4±0.9 | 36.4 |
| | Multiple | SHOT-ens | **33.3±2.1** | 32.1±1.8 | 34.1±1.9 | 36.5±1.2 | 38.1±1.4 | 39.9±0.9 | 40.4±0.9 | 36.3 |
| | | TOHAN-ens | 31.7±1.8 | 31.0±1.4 | 35.8±1.3 | 36.9±0.9 | **40.5±0.6** | 42.6±0.8 | 43.1±0.7 | 37.4 |
| | | DECISION | 30.3±2.1 | 30.5±2.0 | 31.2±1.8 | 31.5±1.9 | 32.0±1.2 | 32.1±1.3 | 32.4±0.9 | 31.4 |
| | | Bi-ATEN | 31.5±1.5 | 30.9±1.9 | 33.5±1.7 | 33.1±1.8 | 35.3±1.6 | 35.3±0.8 | 37.1±1.4 | 33.8 |
| | | HiFE (ours) | 33.0±2.0 | **32.9±1.2** | **37.5±0.8** | **39.8±0.8** | 40.1±1.0 | **42.7±1.1** | **43.3±0.9** | **39.5** |

Table 2: Classification accuracy±standard deviation (%) on three adaptation tasks of **digit** datasets. M, U, and S refer to MNIST, USPS, and SVHN, respectively. The suffixes -best and -worst refer to the best and worst results after adapting each single source hypothesis. The suffixes -strong and -ens refer to the result of adapting the strong hypothesis and the ensemble of all hypotheses, respectively. Results of SHOT (Liang et al., 2020), TOHAN (Chi et al., 2021), DECISION (Ahmed et al., 2021), Bi-ATEN (Li et al., 2024), and our HiFE are presented. The highest accuracy is marked in bold.

with office datasets in Table 3 (more results can be found in Appendix E). The proposed HiFE consistently improves the adaptation performance, boosting the average accuracy from 60.1% to 64.2% in task $W \to A$. While HiFE is designed to adapt source models from a single SD to a TD, our approach also works effectively when the source models come from multiple domains (see Appendix F). In addition, HiFE also outperforms the SOTA in the partial FHA scenario (as shown in Appendix G).

**Results of image classification tasks**. For image classification, we evaluate our approach on two adaptation tasks, $CF \to ST$ and $ST \to CF$. As shown in Table 4, we achieve 1.3% average improvement over the SOTA ensemble approaches in the $CF \to ST$ task.

**Results of VisDA-C object classification tasks**. Table Section 5.2 presents the results of the synthesis-to-real adaptation task using the large-scale VisDA-C dataset. When the number of target samples is limited to 10 per class, SHOT-best achieves a significantly higher accuracy (74.8%) compared to SHOT-strong (70.8%). This observation supports our hypothesis that certain weak hypotheses can outperform strong hypotheses after adaptation. As the number of target samples increases to 50 per class, the performance gap between SHOT-strong and SHOT-best narrows, suggesting that the dependency on the source model diminishes with more target data. A detailed comparison of HiFE with previous methods reveals that HiFE excels, particularly when fewer samples are available. For example, when the number of target samples per class is 10, HIFE (76.1%) outperforms the SOTA methods Bi-ATEN and TOHAN (72.5%) by 3.6%. This finding indicates that HiFE effectively leverages weak hypotheses to enhance adaptation accuracy under conditions of data scarcity.

| Tasks | Hypothesis Number | Methods | Number of Target Data per Class | | | | | Average |
|---|---|---|---|---|---|---|---|---|
| | | | 1 | 2 | 3 | 4 | 5 | |
| $A \to D$ | Multiple | SHOT-ens | 76.5±1.9 | 78.6±1.5 | 78.7±1.7 | 80.1±0.9 | 81.2±1.2 | 79.0 |
| | | TOHAN-ens | 78.5±1.6 | 79.5±1.3 | 83.2±0.9 | 85.1±1.1 | 87.1±1.1 | 82.7 |
| | | DECISION | **79.2±1.5** | 80.2±2.2 | 80.8±1.2 | 81.5±2.0 | 83.5±1.3 | 81.0 |
| | | Bi-ATEN | 79.1±1.2 | 81.6±1.4 | 81.5±0.9 | 82.1±1.5 | 84.1±2.1 | 81.7 |
| | | HiFE (ours) | **79.2±1.0** | **84.3±0.4** | **85.7±1.0** | **86.2±0.9** | **89.2±0.8** | **85.0** |
| $D \to A$ | Multiple | SHOT-ens | 56.8±2.0 | 58.0±1.9 | 59.2±1.7 | 61.8±0.5 | 62.5±0.9 | 59.7 |
| | | TOHAN-ens | 58.1±1.3 | 60.8±1.2 | 63.1±1.9 | 63.8±0.8 | 64.1±0.9 | 62.0 |
| | | DECISION | 54.1±1.6 | 54.2±2.5 | 56.1±2.1 | 57.4±0.9 | 58.5±0.7 | 56.1 |
| | | Bi-ATEN | 55.2±1.3 | 57.1±2.1 | 60.3±0.9 | 61.5±1.1 | 62.5±1.8 | 59.3 |
| | | HiFE (ours) | **61.8±1.0** | **64.7±0.7** | **67.2±0.6** | **66.8±1.0** | **67.5±0.9** | **65.6** |
| $W \to A$ | Multiple | SHOT-ens | 55.1±1.2 | 58.2±1.6 | 59.9±1.4 | 60.8±1.1 | 61.2±1.1 | 59.0 |
| | | TOHAN-ens | 56.5±1.0 | 60.1±0.9 | 60.4±1.2 | 61.2±0.8 | 62.5±0.7 | 60.1 |
| | | DECISION | 54.1±2.1 | 54.9±1.8 | 55.6±1.6 | 56.5±1.2 | 58.1±1.2 | 55.8 |
| | | Bi-ATEN | 56.2±1.9 | 58.4±1.6 | 58.9±2.2 | 61.5±1.3 | 61.7±1.1 | 59.3 |
| | | HiFE (ours) | **62.5±2.5** | **65.1±1.5** | **64.4±1.2** | **64.3±0.9** | **64.8±0.8** | **64.2** |

Table 3: Classification accuracy±standard deviation (%) on three adaptation tasks of office datasets. A, D, and W are abbreviations of Amazon, DSLR, and Webcam. The suffix -ens refers to the result of the ensemble of all adapted hypotheses. Results of SHOT (Liang et al., 2020), TOHAN (Chi et al., 2021), DECISION (Ahmed et al., 2021), Bi-ATEN (Li et al., 2024), and our HiFE are presented. The bold value represents the highest accuracy.

| Tasks | Methods | Number of Target Data per Class | | Average |
|---|---|---|---|---|
| | | 15 | 20 | |
| $CF \to ST$ | SHOT-ens | 70.3±0.4 | 70.5±0.6 | 70.4 |
| | TOHAN-ens | 67.5±0.6 | 69.8±0.5 | 68.7 |
| | DECISION | 70.4±0.4 | 70.6±0.6 | 70.5 |
| | Bi-ATEN | 70.7±0.3 | 70.9±0.5 | 70.8 |
| | HiFE (ours) | **71.6±0.4** | **71.9±0.3** | **71.8** |
| $ST \to CF$ | SHOT-ens | 53.1±0.6 | 53.5±0.5 | 53.3 |
| | TOHAN-ens | 52.5±0.6 | 52.6±0.8 | 52.6 |
| | DECISION | 54.2±0.5 | 54.5±0.6 | 54.4 |
| | Bi-ATEN | 53.7±0.4 | 54.6±0.4 | 54.1 |
| | HiFE (ours) | **55.0±0.3** | **55.3±0.3** | **55.2** |

Table 4: Classification Accuracy±standard deviation (%) on two tasks between CIFAR-10 (CF) and STL-10 (ST).

## 5.3 Ablation Studies

**Ablation study on the feature DeCL loss**. We study the advantage of our training loss by incorporating feature DeCL loss $\mathcal{L}_{\text{DeCL}}$ in Equation (4) with different $\beta$ values ranging from 0 to 1.0 with the digit datasets. In this context, $\beta = 0$ corresponds to training the network using only the supervised loss $\mathcal{L}_{\text{CLS}}$, while $\beta = 1$ corresponds to training the network using only the feature DeCL loss $\mathcal{L}_{\text{DeCL}}$. As shown in Table 6, our optimal results generally occur at $\beta = 0.1$, which yields an average improvement of 1.4% compared to the result obtained when no feature DeCL loss is used ($\beta = 0$). Notably, even when the model is trained solely using the feature DeCL loss $\mathcal{L}_{\text{DeCL}}$ ($\beta = 1.0$), it still achieves an average improvement of 10.9% compared to the accuracy before the adaptation (WA), demonstrating the effectiveness of the feature DeCL loss.

We also visualize the correlation matrices of the features after the merge at the first layer of the task $U \to M$ when $\beta = 0.1$. As depicted in Figure 3, as the training progresses, the feature DeCL loss guides the decrease of most of the correlation values between the four features, thereby increasing feature diversity.

**Ablation study on the number of weak hypotheses**. We conduct an ablation study to analyze the impact of the number of weak hypotheses on the final performance, providing insights into choosing a proper number of weak hypotheses to balance the cost and performance. We do this experiment in an adaptation task from domain Mnist to USPS. We select varying numbers (from 2 to 6) of source hypotheses from the models provided in Section 5.1 and make sure the accuracy range of each group of source models is the same. For each experiment, the selected weak hypotheses started from $h_5$ and ended with $h_{11}$, ensuring the

| Tasks | Hypothesis Number | Methods | Number of Target Data per Class | | | Average |
|---|---|---|---|---|---|---|
| | | | 10 | 30 | 50 | |
| *Synthesis → Real* | Single | SHOT-worst | 70.8±1.2 | 76.1±1.3 | 80.4±0.6 | 75.8 |
| | | SHOT-best | 74.8±0.8 | 78.8±1.1 | 81.3± 0.8 | 78.3 |
| | | SHOT-strong | 70.8±1.3 | 77.7±1.7 | 81.1±0.4 | 76.5 |
| | Multiple | SHOT-ens | 72.1± | 77.1± | 80.9± | 76.7 |
| | | TOHAN-ens | 72.5±1.5 | 77.7±1.2 | 81.5±0.9 | 77.2 |
| | | DECISION | 71.2±2.1 | 76.5±1.9 | 79.8±1.1 | 75.8 |
| | | Bi-ATEN | 72.5±2.0 | 77.7±1.9 | 81.5±0.8 | 77.2 |
| | | HiFE (ours) | **76.1±1.1** | **80.3±0.8** | **82.1±0.7** | **79.5** |

Table 5: Classification Accuracy±standard deviation (%) on the adaptation task from VisDA-C synthesis data to real object data.

| Tasks | The Value of $\beta$ | | | | | | | | WA |
|---|---|---|---|---|---|---|---|---|---|
| | 0 | 0.1 | 0.2 | 0.3 | 0.5 | 0.7 | 0.9 | 1.0 | |
| $S \to M$ | 93.2 | **94.0** | 93.9 | 93.2 | 92.3 | 90.0 | 86.4 | 77.0 | 67.1 |
| $S \to U$ | 94.5 | **95.6** | **95.6** | 94.1 | 92.8 | 91.2 | 88.7 | 82.3 | 78.2 |
| $M \to S$ | 52.2 | 54.2 | **54.3** | 53.2 | 52.8 | 52.7 | 52.1 | 46.0 | 23.2 |
| $M \to U$ | 96.1 | **97.1** | 96.8 | 96.3 | 96.1 | 95.1 | 94.6 | 91.2 | 70.5 |
| $U \to S$ | 44.5 | **46.2** | 44.5 | 42.5 | 40.1 | 39.6 | 39.1 | 33.2 | 26.2 |
| $U \to M$ | 94.9 | **95.9** | 95.3 | 95.1 | 94.6 | 92.9 | 91.3 | 89.9 | 88.5 |
| Average | 79.2 | **80.6** | 80.1 | 79.1 | 78.1 | 76.9 | 75.4 | 69.9 | 59 |

Table 6: Ablation study on the feature decrrelation learning loss balance parameter $\beta$ in Equation (4). M, U, and S are abbreviations of MNIST, USPS, and SVHN. WA indicates the accuracy of the model without the adaptation. The bold value represents the highest accuracy (%).

selected accuracy range was [60, 95). The results are presented in $\text{Exp}_{1-1} \sim \text{Exp}_{1-5}$ of Table 7. As shown in Table 7, when the number of weak hypotheses is less than 3, the average accuracy (94.2% in $\text{Exp}_{1-1}$) is lower than that when the number of weak hypotheses is greater than 3. Moreover, the average accuracy remains consistent when the number of weak hypotheses exceeds 3 (see the comparison of the result of $\text{Exp}_{1-3} \sim \text{Exp}_{1-5}$).

**Ablation study on the accuracy range of weak hypotheses**. To investigate the impact of the accuracy range of weak hypotheses on the final performance, we leverage the source models presented in Section 5.1 and do the digit adaptation task from Mnist to USPS using source models with varying accuracy ranges. We conducted experiments from $\text{Exp}_{2-1}$ to $\text{Exp}_{2-4}$ as outlined in Table 7. Our results indicate that as the accuracy range increases, the performance after adaptation improves. It is important to note that weak source models with an accuracy lower than 55% may harm the final performance. Our comparison of $\text{Exp}_{2-1}$ and $\text{Exp}_{2-5}$ revealed that using weak hypotheses with such low accuracy resulted in worse performance than

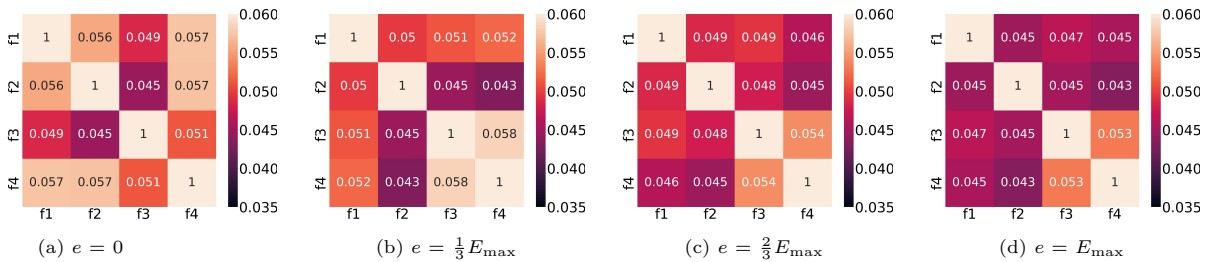

(a) $e = 0$  (b) $e = \frac{1}{3}E_{\max}$  (c) $e = \frac{2}{3}E_{\max}$  (d) $e = E_{\max}$

Figure 3: The correlation matrixes of the features after the merge at the first layer on digit task $U \to M$ when the $\beta$ value is set to 0.1. (a) $\sim$ (d) shows the results over different training stages with $e$, $E_{\max}$ being the current and maximum number of epochs. As the training continues, we observe that most of the correlation values between the four features in this layer decrease (the darker the color, the lower the corresponding correlation value).

| Exp ID | HN | Weak Hypothesis Accuracy Range | Weak Hypothesis Indices | FN | LN | UH | Number of Target Data per Class | | | | Average |
|--------|-----|------------------|------------------|-----|-----|-----|------|------|------|------|---------|
| | | | | | | | 1 | 3 | 5 | 7 | |
| $Exp_{1-1}$ | 2 | $[60, 95)$ | $h_5, h_{11}$ | 2 | 3 | $h_{12}$ | 92.0±1.2 | 94.6±0.7 | 94.5±0.5 | 95.6±0.5 | 94.2 |
| $Exp_{1-2}$ | 3 | $[60, 95)$ | $h_5, h_8, h_{11}$ | 2 | 3 | $h_{12}$ | 92.3±1.0 | 94.8±0.6 | 95.7±0.7 | 96.1±0.7 | 94.7 |
| $Exp_{1-3}$ | 4 | $[60, 95)$ | $h_5, h_7, h_9, h_{11}$ | 2 | 3 | $h_{12}$ | 92.7±1.1 | 95.2±0.6 | 96.1±0.5 | 96.7±0.2 | 95.2 |
| $Exp_{1-4}$ | 5 | $[60, 95)$ | $h_5, h_7, h_8, h_9, h_{11}$ | 2 | 3 | $h_{12}$ | 92.6±1.0 | 95.3±0.7 | 96.0±0.5 | 96.6±0.4 | 95.1 |
| $Exp_{1-5}$ | 6 | $[60, 95)$ | $h_5, h_6, h_7, h_8, h_9, h_{11}$ | 2 | 3 | $h_{12}$ | 92.8±0.9 | 95.1±0.5 | 96.2±0.7 | 96.8±0.5 | 95.2 |
| $Exp_{2-1}$ | 3 | $[40, 55)$ | $h_1, h_2, h_3$ | 2 | 3 | $h_{12}$ | 91.1±0.7 | 93.8±0.6 | 94.5±0.8 | 95.2±0.8 | 93.7 |
| $Exp_{2-2}$ | 3 | $[55, 70)$ | $h_4, h_5, h_6$ | 2 | 3 | $h_{12}$ | 91.5±0.7 | 94.1±0.6 | 95.3±0.6 | 96.0±0.8 | 94.2 |
| $Exp_{2-3}$ | 3 | $[70, 85)$ | $h_7, h_8, h_9$ | 2 | 3 | $h_{12}$ | 92.2±0.6 | 94.7±0.8 | 95.7±0.8 | 96.7±0.6 | 94.8 |
| $Exp_{2-4}$ | 3 | $[80, 95)$ | $h_9, h_{10}, h_{11}$ | 2 | 3 | $h_{12}$ | 92.4±0.8 | 94.8±0.4 | 96.1±0.8 | 96.7±0.6 | 95.0 |
| $Exp_{2-5}$ | 0 | - | - | 1 | 1 | $h_{12}$ | 91.5±0.6 | 94.2±0.6 | 94.2±0.9 | 95.5±0.7 | 93.9 |
| $Exp_{3-1}$ | 7 | $[60, 95)$ | $h_{5\sim11}$ | 2 | 4 | $h_{12}$ | 93.0±1.4 | 95.3±0.5 | 96.0±0.3 | 96.7±0.3 | 95.3 |
| $Exp_{3-2}$ | 7 | $[60, 95)$ | $h_{5\sim11}$ | 4 | 3 | $h_{12}$ | 93.2±1.1 | 94.9±0.6 | 96.2±0.5 | 96.6±0.4 | 95.2 |
| $Exp_{3-3}$ | 7 | $[60, 95)$ | $h_{5\sim11}$ | 8 | 1 | $h_{12}$ | 92.3±1.2 | 93.1±1.0 | 95.4±0.8 | 95.6±0.5 | 94.1 |
| $Exp_{3-4}$ | 7 | $[60, 95)$ | $h_{5\sim11}$ | / | 1 | $h_{12}$ | 92.1±1.0 | 92.5±0.9 | 94.8±0.6 | 95.1±0.7 | 93.6 |
| $Exp_{4-1}$ | 8 | $[60, 95)$ | $h_{5\sim11}$ | 2 | 8 | $h_{12}$ | 93.0±0.9 | 95.3±0.5 | 96.0±0.3 | 96.7±0.3 | 95.3 |
| $Exp_{4-2}$ | 8 | $[60, 95)$ | $h_{5\sim11}$ | 2 | 8 | $h_{11,12}$ | 93.2±0.7 | 95.6±0.7 | 95.8±0.8 | 96.9±0.5 | 95.4 |
| $Exp_{4-3}$ | 8 | $[60, 95)$ | $h_{5\sim11}$ | 2 | 8 | $h_{10\sim12}$ | 92.5±0.6 | 95.1±0.7 | 95.4±0.5 | 96.1±0.4 | 94.8 |
| $Exp_{4-4}$ | 8 | $[60, 95)$ | $h_{5\sim11}$ | 2 | 8 | $h_{5\sim12}$ | 91.0±0.8 | 93.1±0.6 | 93.8±0.6 | 94.1±0.3 | 93.0 |
| $Exp_{4-5}$ | 8 | $[60, 95)$ | $h_{5\sim11}$ | 2 | 8 | $h_{5\sim10}$ | 88.1±0.9 | 89.1±0.8 | 89.5± 1.1 | 89.7±0.7 | 89.1 |

Table 7: Classification accuracy±standard deviation (%) on digit adaptation task $Mnist \rightarrow USPS$ with varying parameters including the number of weak hypotheses (HN), the accuracy ranges of the weak hypotheses, the number of input features to each WRU (FN), the layer number (LN) in the hierarchical feature ensemble module and the updated hypotheses during fine-tuning (UH). $h_{12}$ is the strong hypothesis.

adaptation without weak hypotheses. To summarize, our approach, HiFE, performs well when incorporating a larger number of weak hypotheses, provided that these hypotheses are not of very low accuracy. In our experiments, we selected 7 weak hypotheses as our base setting to balance performance and computational cost. In practical applications, the selection of weak hypotheses can be guided by the available computational resources and the performance of each weak hypothesis on the SD.

**Ablation study on the hierarchical layer number.** To investigate the impact of the layer number in the ensemble module on the final performance, we have experimented with varying numbers of input features fed into WRU, which subsequently alters the number of merge layers within the hierarchical feature ensemble module. we conducted experiments from $Exp_{3-1}$ to $Exp_{3-4}$, modifying the number of input features for each WRU. The results are presented in Table 7. Notably, in $Exp_{3-4}$, we utilized a simple weighted feature sum to merge all source features rather than using HEFM with WRU for feature aggregation. When we set the number of input features for each WRU to 2 or 4, with the corresponding layer number in the ensemble module to be 4 and 3, respectively, the adaptation average accuracy is similar. However, when the number of input features for each WRU increases to 8, we apply one WRU to merge the eight source encoders at once, and the adaptation average accuracy decreases to 94.1%.

**Ablation study on the updated hypotheses.** In our HiFE approach, we aggregate multiple hypotheses. During the fine-tuning stage, we fix all the weak hypotheses and only update the parameters in the strong hypotheses. We conducted experiments from $Exp_{4-1}$ to $Exp_{4-4}$, modifying which hypotheses are updated during the fine-tuning stage. $Exp_{4-1}$ is our default setting, where only the last strong hypothesis $h_{12}$ is updated and it achieves good performance. When we update all the weak and strong hypotheses together ($Exp_{4-3}$), or only update all the weak hypotheses ($Exp_{4-4}$), the performance deteriorates. Notably, when we update the last two models $h_{11}$ and $h_{12}$ ($Exp_{4-2}$), we achieve the best performance. This is because $h_{12}$ tends to overfit to the SD while $h_{11}$ is better suited for the TD (see the comparison of the result of adaptation of $h_{11}$ (SHOT-best) and $h_{12}$ (SHOT-strong) of task $M \rightarrow U$ in Table 8 in Appendix Sec D). In practice, the few-shot target samples are insufficient to guide the fine-tuning of both weak and strong hypotheses together. Additionally, it is unrealistic to determine which source hypothesis is best for the target domain beforehand. Therefore, the most suitable approach is to only update the final strong hypotheses while keeping all other weak hypotheses unchanged among the source encoders. By designing and updating the hierarchical feature ensemble module, we can merge all these models to obtain the best final model.

# 6 Limitations and Future Work

HiFE leverages multiple source hypotheses with varying accuracy levels from the SD to improve the performance of models in the TD. By exploiting the diversity of source models, HiFE has the potential to enhance the generalization capabilities of the adapted models. However, the additional hypotheses result in increased model transfer and storage costs. Moreover, the increase in the number of parameters of the target model leads to higher computational costs. Nonetheless, we argue that the benefits of leveraging multiple source models with different strengths outweigh the costs associated with processing additional hypotheses, particularly when source data is absent for transfer. With the growing need to address privacy concerns and mitigate data-sharing challenges in real-world applications, opting for weak models simplifies collaboration between source providers and users.

For future research, it would be beneficial to investigate methods for generating weak hypotheses with higher diversity. Although the current experiments obtained weak hypotheses in the same run as generating the final strong hypotheses for simplicity, there is potential for improvement by obtaining weak hypotheses through different random seeds, hyperparameter choices, or training on different subsets of the source data. By increasing the diversity of weak hypotheses, we could obtain better performance after adaptation and further improve the effectiveness of the proposed approach.

# 7 Conclusion

In this paper, we investigate the potential of utilizing weak source hypotheses for domain adaptation and introduce a new problem setting termed "few-shot hypotheses adaptation with weak models". To tackle this problem, we design a new framework called HiFE, which leverages an array of readily available weak hypotheses to improve the adaptation performance of a strong source hypothesis. As a result, HiFE significantly mitigates the occurrence of over-fitting under the few-shot setting and achieves the SOTA performance across various adaptation tasks. This research introduces an innovative perspective for addressing the FHA problem in scenarios where the source data is inaccessible and the target data is limited. Additionally, this research shed light on the use of a weak source model to boost the practical application of transfer learning in scenarios where data privacy concerns are on the rise.

# 8 Acknowledgments

We would like to express our sincere gratitude to the anonymous reviewers for their valuable feedback and constructive suggestions. Their insightful comments greatly contributed to improving the quality and clarity of this paper.

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

## Supplementary Material for "HiFE: Hierarchical Feature Ensemble Framework for Few-shot Hypotheses Adaptation"

We organize the appendix as follows:

- In A, we provide a pilot study on the FHA process from SVHN to Mnist data.

- In B, we provide some implementation details of the experiments.

- In C, we provide a brief overview of the related theorems that support the proposed approach and highlight their relevance to the work presented in the paper.

- In D, we present additional results of experiments on digit datasets to supplement and reinforce the findings discussed in the main text.

- In E, we provide more results of the experiments on office datasets.

- In F, we apply HiFE to a multi-source multi-model adaptation scenario and analyze the performance.

- In G, we extend HiFE to a partial few-shot hypothesis adaptation scenario, exposing the model's versatility and potential for future applications.

- In H, we provide some analyses on the model complexity.

## A    Pilot Study on the FHA Process From SVHN to Mnist.

The strong source model may overfit the SD and perform worse on the TD after the adaptation. To justify our assumption, we empirically design an experimental task that adapts source models with varying source accuracies from the digit dataset SVHN to the target dataset Mnist. Figure 4 shows that the weak source hypotheses (e.g., Model-4 [source acc=76.0%]) could perform better than the strong source models (e.g., Model-1 [source acc = 92.2%]) on the TD, indicating that weak hypotheses can generalize better on the TD.

## B    Implementation Details

**Training target hypotheses with HiFE.** As we mentioned in the manuscript Section 5, we generate 12 source models $\{h_i|_{i=1}^{12}\}$. Among these models, we used $\{h_i|_{i=5}^{12}\}$ (8 models) in the experiments shown in Section 5.2. In our HiFE, we set the number of input features for each WRU to two, leading to a four-layer feature ensemble structure.

The network uses the PyTorch framework on a PC with four NVIDIA 2080ti GPUs. We trained the source hypothesis using a stochastic gradient descent (SGD) optimizer with a momentum value of 0.5 with the learning rate initialized to 1e-2 and decreased to 1e-5 step by step. During the adaptation, we adopt SGD with Nesterov momentum (Ruder, 2016) with a momentum value of 0.9. Following (Liang et al., 2020), we insert a batch normalization layer and a weight normalization layer before the end of each encoder and classifier, respectively.

## C    Review of the Theorems

In this section, we present a brief overview of the related theorems of Theorem 1, which is derived under the PAC-Bayes framework (McAllester, 1999). The PAC-Bayes theory provides data-dependent generalization bounds over the generalization error $\mathbb{E}_{\rho(\theta)}(L(\theta))$ of a model with parameter $\theta$ under $i.i.d.$ data. In recent years, a PAC-Bayes bound (Germain et al., 2009) has been widely adopted because it can apply to general unbounded losses (e.g., log-loss). We restate here this PAC-Bayes bound as follows:

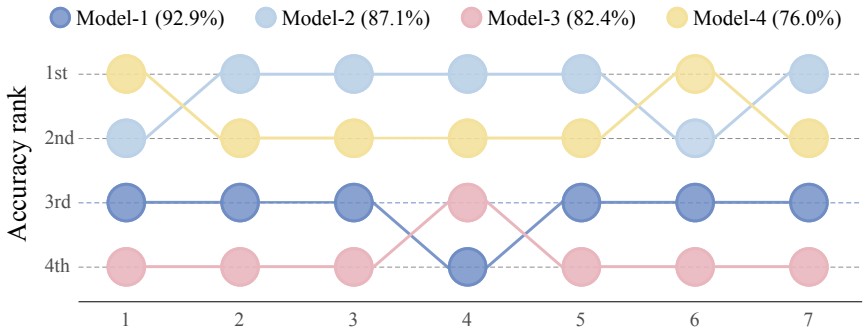

Figure 4: This figure depicts the FHA process from SVHN to Mnist. Four source models were generated using the training data from SVHN and fine-tuned with different quantities of samples from Mnist. The y-axis indicates the performance rank after adaptation, with the highest accuracy in the TD ranked first. Model-4, despite having the lowest source accuracy (76%), exhibits notably superior performance on the TD compared to Model-1, which has the highest source accuracy (92.9%).

**Theorem 2.** *(PAC-Bayes bounds (Germain et al., 2009; Masegosa, 2020; McAllester, 1999)). Given a data distribution $P$ over $\mathcal{X} \times \mathcal{Y}$, a hypothesis space $\Theta$, a prior distribution $\pi$ over $\Theta$, for any $\delta \in (0, 1]$ and $\lambda > 0$, with probability at least $1 - \delta$ over samples $D \sim P^n$, we have for all posterior $\rho$,*

$$\mathbb{E}_{\rho(\theta)}(L(\theta) \leq \mathbb{E}_{\rho(\theta)}[\hat{L}(\theta, D)] + \frac{1}{\lambda}[D_{KL}(\rho \parallel \pi) + \ln\frac{1}{\delta} + \Psi_{P,\pi}(\lambda, n)],$$

*where $\Psi_{P,\pi}(\lambda, n) = \ln\mathbb{E}_{\pi(\theta)}\mathbb{E}_{D\sim P^n}[e^{\lambda(L(\theta)-\hat{L}(\theta,D))}]$.*

Based on the second-order Jensen inequalities ((Needham, 1993)), the second-order oracle bound with tighter PAC-Bayes bounds is derived. (Deng et al., 2023) further extends this theory to ensemble models as shown in Theorem 3. (Deng et al., 2023) also shows a second-order PAC-Bayesian bound over the performance of the posterior predictive distribution of the averaging ensemble model, which is shown in Theorem 4.

**Theorem 3.** *(Second-order oracle bound ((Deng et al., 2023; Masegosa, 2020; Needham, 1993))). Given a data distribution $P$, a set of model parameters $\{\theta_i\}_{i=1}^M$, for any distribution $\{\rho_i\}_{i=1}^M$ over $\{\Theta_i\}_{i=1}^M$ satisfies that*

$$\mathbb{E}_{\rho(\theta)}(L(\theta) \leq \frac{1}{M}\sum_{i=1}^M \mathbb{E}_{\rho_i(\theta)}[L(\theta)] - \mathbb{V}(\rho(\theta)),$$

*where $\theta_i \in \Theta_i$, $\theta = \{\theta_i\}_{i=1}^M$ and $\rho(\theta) = \prod_{i=1}^M \rho_i(\theta_i)$ and $\mathbb{V}(\rho(\theta))$ is a variance term defined as*

$$\mathbb{V}(\rho(\theta)) = \mathbb{E}_{\rho(\theta)}\mathbb{E}_{(x,y)\sim P}\left[\frac{1}{2M\max_\theta p(y|x,\theta)^2}\sum_{i=1}^M\left(p(y|x,\theta_i) - \frac{1}{M}\sum_{k=1}^M p(y|x,\theta_k)\right)^2\right].$$

**Lemma 1.** *For any distribution $\{\rho_i\}_{i=1}^M$ over $\{\Theta_i\}_{i=1}^M$, the Second-order Jensen bound of Theorem 3 can be expressed as follows (Deng et al., 2023) ,*

$$\frac{1}{M}\sum_{i=1}^M \mathbb{E}_{\rho_i(\theta_i)}[L(\theta_i)] - \mathbb{V}(\rho(\theta)) = \mathbb{E}_{\rho(\theta)}L_2(\theta),$$

*where*

$$L_2(\theta) = \mathbb{E}_{(x,y)\sim P}\left[\frac{1}{M}\sum_{i=1}^{M}\left(\log(y|x,\theta_i) - \frac{\left(p(y|x,\theta) - \frac{1}{M}\sum_{i=1}^{M}p(y|x,\theta_k)\right)^2}{2\max_\theta p(y|x,\theta)^2}\right)\right].$$

**Theorem 4.** *(Model ensemble error bound ((Deng et al., 2023))). Given a data distribution $P$ over $\mathcal{X} \times \mathcal{Y}$, a set of model parameters $\{\Theta_i\}_{i=1}^{M}$ and associated prior $\{\pi_i\}_{i=1}^{M}$, where $\pi_i$ is defined over $\Theta_i$, a $\delta \in (0,1]$, and a real number $c > 0$, with probability at least $1 - \delta$ over samples $D \sim P^n$, we have for all posterior $\{\rho_i\}_{i=1}^{M}$ over $\{\Theta_i\}_{i=1}^{M}$,*

$$\mathbb{E}_{\rho(\theta)}(L(\theta) \le \frac{1}{M}\sum_{i=1}^{M}\left(\mathbb{E}_{\rho_i(\theta_i)}[\hat{L}(\theta_i', D)] + \frac{D_{KL}(\rho_i \parallel \pi_i)}{cn}\right) - \hat{\mathbb{V}}(\rho(\theta), D) + \frac{\epsilon}{cnL},$$

*where $\hat{\mathbb{V}}(\rho(\theta), D)$ is the empirical version of a variance term $\mathbb{V}(\rho(\theta))$ and $\epsilon$ is defined as*

$$\epsilon(P, \pi, c, n, \delta) = \log\mathbb{E}_{\pi(\theta)}\mathbb{E}_{D\sim P^n}\left[e^{cn\left(\sum_{i=1}^{M}\left(L(\theta_i) - \hat{L}(\theta_i, D)\right) - M\left(\mathbb{V}(\theta) - \hat{\mathbb{V}}(\theta, D)\right)\right)}\right] + \log\frac{1}{\delta}.$$

**Lemma 2.** *If there exists an input sample $x \in P_\mathcal{X}$ such that $h_{\theta_i}(x) \ne h_{\theta_j}(x)$, we then have $\mathbb{V}(\rho(\theta)) > 0$ ((Deng et al., 2023)).*

Our method proposes to add multiple weak hypotheses for training, thereby improving the diversity of the source model. From the above theorem, our approach intends to look for a tighter upper bound of the expected loss.

To further analyze the term $D_{KL}(\rho_i \parallel \pi_i)$ in Theorem 4 when providing $n$ training samples from the target domain, we assume $\theta_i \in \mathbb{R}^{d_i}$, $\pi_i(\theta_i) \sim \mathcal{N}(0, \sigma^2 I)$, and $\rho_i(\theta_i)$ is a Dirac-delta distribution centered around $\theta_i'$ with $\rho_i(\theta_i) = \delta_{\theta_i'}(\theta_i)$, $\forall i \in [M]$, then we have

$$\mathbb{E}_{\rho_i(\theta_i)}[\hat{L}(\theta_i, D)] = \mathbb{E}_{\delta_{\theta_i'}(\theta_i)}[\hat{L}(\theta_i, D)] = \int \delta_{\theta_i'}\hat{L}(\theta_i, D)d\theta_i = \hat{L}(\theta_i', D),$$

and

$$D_{KL}(\rho_i \parallel \pi_i) = \int \delta_{\theta_i'}\log\frac{\delta_{\theta_i'}(\theta_i)}{\pi(\theta_i)}d\theta_i = -\log\pi\theta_i' = \frac{d_i}{2}log(2\pi\sigma^2) + \frac{1}{2\sigma^2}\parallel \pi_i' \parallel^2.$$

Hence, the equation in Theorem 4 can be updated to

$$\mathbb{E}_{\rho(\theta)}(L(\theta) \le \frac{1}{M}\sum_{i=1}^{M}\left(\hat{L}(\theta_i', D) + \frac{1}{2cn\sigma^2}\parallel \theta_i \parallel^2 + \frac{d_i}{2cn}\log(2\pi\sigma^2)\right) - \hat{\mathbb{V}}(\rho(\theta), D) + \frac{\epsilon}{cnL}.$$

This error bound is what we showed in Theorem 1 in the manuscript.

## D    More Results on Digit Datasets

We evaluate our approach on six closed-set adaptation tasks on digit datasets. We report the results of $M \to S$, $S \to U$, and $M \to U$ in Table 8. Our HiFE achieves 3.7%, 2.1%, and 2.3% average improvement over the SOTA on these three tasks, respectively.

| Tasks | Hypothesis Number | Methods | Number of Target Data per Class | | | | | | | Average |
|---|---|---|---|---|---|---|---|---|---|---|
| | | | 1 | 2 | 3 | 4 | 5 | 6 | 7 | |
| $M \to S$ | Single | SHOT-worst | 16.3±1.7 | 15.1±1.5 | 16.6±1.1 | 17.3±1.2 | 16.7±2.1 | 16.8±1.9 | 17.5±0.9 | 16.6 |
| | | SHOT-best | 28.2±1.2 | 33.5±1.8 | 35.3±2.1 | 36.9±2.3 | 43.0±0.9 | 47.2±2.1 | 50.1±1.5 | 39.2 |
| | | SHOT-strong | 28.0±1.7 | 33.0±1.5 | 35.0±1.3 | 36.2±2.0 | 39.0±1.8 | 47.0±0.8 | 50.0±0.7 | 38.3 |
| | Multiple | SHOT-ens | 28.2±1.8 | 29.9±1.6 | 33.4±1.4 | 34.2±1.2 | 34.6±1.3 | 35.7±0.9 | 37.3±0.8 | 33.3 |
| | | TOHAN-ens | 30.2±1.2 | **41.0±1.5** | 41.1±1.3 | 41.4±0.9 | 42.8±1.1 | 43.1±1.2 | 45.5±1.1 | 40.1 |
| | | DECISION | 24.2±0.9 | 26.9±1.6 | 27.6±1.1 | 29.4±1.3 | 30.9±0.5 | 33.2±0.8 | 35.3±0.9 | 29.6 |
| | | Bi-ATEN | 28.5±0.9 | 31.1±1.2 | 32.1±0.8 | 35.5±1.7 | 38.1±1.5 | 38.5±1.9 | 39.1±1.4 | 34.7 |
| | | HiFE (ours) | **30.4±1.4** | 40.8±1.1 | **43.5±1.2** | **45.7±0.5** | **46.9±1.9** | **48.8±1.2** | **50.3±0.9** | **43.8** |
| $S \to U$ | Single | SHOT-worst | 44.4±1.4 | 45.5±1.1 | 51.2±0.9 | 50.3±1.5 | 50.8±1.9 | 50.8±0.7 | 50.6±0.6 | 49.1 |
| | | SHOT-best | 82.9±0.8 | 81.4±0.7 | 88.3±1.2 | 86.9±1.8 | 87.5±0.8 | 87.1±0.7 | 88.1±0.6 | 86.0 |
| | | SHOT-strong | 81.1±1.6 | 80.8±1.4 | 85.2±0.9 | 85.3±1.1 | 85.2±1.8 | 85.0±1.7 | 85.1±0.5 | 84.0 |
| | Multiple | SHOT-ens | 80.8±2.1 | 81.8±1.6 | 84.6±1.3 | 86.1±1.1 | 86.5±0.9 | 86.2± 0.5 | 87.5±0.4 | 84.8 |
| | | TOHAN-ens | **86.1±1.0** | 89.1±1.5 | 90.0±0.8 | 91.5±0.2 | 92.8±0.5 | 93.4±1.1 | 92.5±1.2 | 90.8 |
| | | DECISION | 73.5±1.2 | 73.7±1.0 | 76.2±0.8 | 78.5±0.8 | 79.7±0.5 | 80.1±0.7 | 82.1±1.0 | 77.7 |
| | | Bi-ATEN | 75.5±2.1 | 78.1±1.6 | 78.9±1.4 | 83.2±3.0 | 84.1±2.3 | 88.2±1.8 | 88.9±1.5 | 82.4 |
| | | HiFE (ours) | **86.1±1.6** | **90.6±0.4** | **93.0±1.2** | **94.0±0.8** | **95.0±0.6** | **96.0±0.2** | **95.9±0.5** | **92.9** |
| $M \to U$ | Single | SHOT-worst | 54.0±2.3 | 54.1±1.7 | 58.3±1.2 | 58.2±1.6 | 58.6±0.7 | 58.8±1.5 | 58.5±1.5 | 57.2 |
| | | SHOT-best | 92.2±2.3 | **94.2±2.3** | 94.6±1.8 | 94.7±1.5 | 94.7±1.8 | 94.7±1.6 | 94.9±1.2 | 94.3 |
| | | SHOT-strong | 92.0±2.4 | 94.1±2.0 | 94.5±1.9 | 94.3±1.6 | 94.3±1.5 | 94.6±1.6 | 94.2±0.9 | 94.0 |
| | Multiple | SHOT-ens | 86.3±1.1 | 86.8±1.5 | 88.8±1.8 | 88.1±1.6 | 88.2±0.4 | 88.0±0.8 | 88.0±0.6 | 87.7 |
| | | TOHAN-ens | 88.4±1.8 | 92.8±1.5 | 93.5±1.6 | 92.8±0.9 | 93.5±0.6 | 94.8±0.8 | 95.1±0.5 | 93.0 |
| | | DECISION | 82.9±2.3 | 82.9±1.0 | 83.3±1.5 | 84.2±1.8 | 85.5±1.3 | 84.6± 0.9 | 85.1±0.8 | 84.1 |
| | | Bi-ATEN | 81.5±1.6 | 84.1±2.3 | 85.6±1.8 | 88.1±1.5 | 87.9±0.9 | 89.7±0.7 | 91.4±1.6 | 86.9 |
| | | HiFE (ours) | **93.0±0.9** | 94.0±1.2 | **95.3±0.5** | **95.5±0.8** | **96.0±0.3** | **96.3±0.2** | **96.7±0.3** | **95.3** |

Table 8: Classification accuracy±standard deviation (%) on two adaptation tasks of digit datasets. M, U, and S refer to Mnist, USPS, and SVHN, respectively. The suffixes of -best and -worst refer to the best and worst results after adapting each single source hypothesis. The suffixes -strong and -ens refer to the result after adapting the strong hypothesis and the ensemble of all adapted hypotheses, respectively. Results of SHOT (Liang et al., 2020), TOHAN (Chi et al., 2021), DECISION (Ahmed et al., 2021), Bi-ATEN (Li et al., 2024), and our HiFE are presented. The highest accuracy is marked in bold.

# E More Results on Office Datasets

The complete results of the closest-set adaptation tasks with office datasets are presented in Table 9. Our analysis reveals that HiFE outperforms SHOT-strong, indicating that incorporating weak hypotheses can lead to improved performance. Regarding multi-hypotheses adaptation results, HiFE achieved average accuracy improvements of 4.1% and 3.6% over the SOTA on $W \to A$ and $D \to A$ tasks, respectively. These results demonstrate that HiFE can circumvent the potentially severe negative transfer induced by weak source hypotheses (as observed in the SHOT-worst result) and outperform previous model ensemble approaches.

# F Experiments on Multi-domain Multi-hypotheses Adaptation Scenario

In our manuscript, HiFE is designed for adapting source models from one single source domain to a target domain. Individuals may wonder whether HiFE still works if the source models come from multiple domains. To answer this question, we do experiments on office datasets (Amazon (A), DSLR (D), Webcam (W)). We train one strong and seven weak hypotheses for each domain in our single-source multi-model adaptation experiment. To adopt models from multiple domains, we randomly choose one strong and seven weak hypotheses among the 16 hypotheses from domains D and W and adapt them to A. As shown in Table 10, HiFE outperforms the SOTA with 4.4% average improvement in this multi-domain and multi-model adaptation task. Moreover, compared with the single-source domain adaptation tasks $D \to A$ and $W \to A$, HiFE achieves 1.0% and 2.4% improvement on the multi-source domain adaptation task $D, W \to A$, respectively.

| Tasks | Hypothesis Number | Methods | Number of Target Data per Class | | | | | Average |
|---|---|---|---|---|---|---|---|---|
| | | | 1 | 2 | 3 | 4 | 5 | |
| $A \to D$ | Single | SHOT-worst | 42.6±2.5 | 45.6±1.8 | 45.3±2.3 | 46.4±1.5 | 46.8±0.9 | 45.3 |
| | | SHOT-best | 78.1±1.8 | 78.6±1.7 | 79.0±2.1 | 80.0±1.8 | 81.2±0.8 | 79.4 |
| | | SHOT-strong | 74.0±2.1 | 76.6±1.5 | 77.4±1.9 | 76.2±0.9 | 81.0±1.2 | 77.1 |
| | Multiple | SHOT-ens | 76.5±1.9 | 78.6±1.5 | 78.7±1.7 | 80.1±0.9 | 81.2±1.2 | 79.0 |
| | | TOHAN-ens | 78.5±1.6 | 79.5±1.3 | 83.2±0.9 | 85.1±1.1 | 87.1±1.1 | 82.7 |
| | | DECISION | **79.2±1.5** | 80.2±2.2 | 80.8±1.2 | 81.5±2.0 | 83.5±1.3 | 81.0 |
| | | Bi-ATEN | 79.1±1.2 | 81.6±1.4 | 81.5±0.9 | 82.1±1.5 | 84.1±2.1 | 81.7 |
| | | HiFE (ours) | **79.2±1.0** | **84.3±0.4** | **85.7±1.0** | **86.2±0.9** | **89.2±0.8** | **85.0** |
| $D \to A$ | Single | SHOT-worst | 17.4±1.9 | 18.8±2.1 | 20.3±1.0 | 20.1±1.5 | 22.3±0.7 | 19.8 |
| | | SHOT-best | 60.2±1.6 | 62.8±2.0 | 63.3±0.8 | **66.8±2.0** | 66.3±1.1 | 63.9 |
| | | SHOT-strong | 60.1±0.9 | 62.5±1.5 | 63.1±1.2 | 63.2±2.1 | 64.1±2.0 | 62.6 |
| | Multiple | SHOT-ens | 56.8±2.0 | 58.0±1.9 | 59.2±1.7 | 61.8±0.5 | 62.5±0.9 | 59.7 |
| | | TOHAN-ens | 58.1±1.3 | 60.8±1.2 | 63.1±1.9 | 63.8±0.8 | 64.1±0.9 | 62.0 |
| | | DECISION | 54.1±1.6 | 54.2±2.5 | 56.1±2.1 | 57.4±0.9 | 58.5±0.7 | 56.1 |
| | | Bi-ATEN | 55.2±1.3 | 57.1±2.1 | 60.3±0.9 | 61.5±1.1 | 62.5±1.8 | 59.3 |
| | | HiFE (ours) | **61.8±1.0** | **64.7±0.7** | **67.2±0.6** | 66.8±1.0 | **67.5±0.9** | **65.6** |
| $A \to W$ | Single | SHOT-worst | 48.1±2.1 | 48.5±1.9 | 48.8±1.5 | 49.1±1.1 | 49.6±0.5 | 48.8 |
| | | SHOT-best | 77.3±1.8 | 77.9±1.2 | 77.9±1.5 | 78.9±0.9 | 79.5±0.5 | 78.3 |
| | | SHOT-strong | 76.2±0.9 | 76.2±0.5 | 75.7±1.9 | 75.2±1.2 | 75.2±0.9 | 75.7 |
| | Multiple | SHOT-ens | 77.6±1.1 | 78.3±0.9 | 78.5±1.3 | 79.5±1.7 | 81.1±1.0 | 79.0 |
| | | TOHAN-ens | 82.5±1.2 | **88.3±2.1** | **88.9±1.0** | 87.1±1.1 | 88.9±0.6 | 87.1 |
| | | DECISION | 82.2±1.1 | 82.8±2.1 | 83.6±2.0 | 84.1±0.9 | 85.6±0.4 | 83.7 |
| | | Bi-ATEN | 81.5±1.7 | 83.5±1.9 | 85.2±1.1 | 87.1±0.9 | 88.5±1.3 | 85.2 |
| | | HiFE (ours) | **84.2±1.3** | 88.0±0.6 | **88.9±0.6** | **90.2±0.5** | **90.3±0.7** | **88.4** |
| $W \to A$ | Single | SHOT-worst | 31.9±1.0 | 32.1±2.1 | 31.8±1.6 | 32.1±1.1 | 33.0±0.9 | 32.2 |
| | | SHOT-best | 61.9±0.8 | 62.0±1.2 | 61.7±1.1 | 61.9±0.7 | 63.1±1.8 | 62.1 |
| | | SHOT-strong | 60.9±1.5 | 60.9±1.1 | 61.6±1.3 | 61.8±0.9 | 62.1±0.8 | 61.5 |
| | Multiple | SHOT-ens | 55.1±1.2 | 58.2±1.6 | 59.9±1.4 | 60.8±1.1 | 61.2±1.1 | 59.0 |
| | | TOHAN-ens | 56.5±1.0 | 60.1±0.9 | 60.4±1.2 | 61.2±0.8 | 62.5±0.7 | 60.1 |
| | | DECISION | 54.1±2.1 | 54.9±1.8 | 55.6±1.6 | 56.5±1.2 | 58.1±1.2 | 55.8 |
| | | Bi-ATEN | 56.2±1.9 | 58.4±1.6 | 58.9±2.2 | 61.5±1.3 | 61.7±1.1 | 59.3 |
| | | HiFE (ours) | **62.5±2.5** | **65.1±1.5** | **64.4±1.2** | **64.3±0.9** | **64.8±0.8** | **64.2** |
| $D \to W$ | Single | SHOT-worst | 48.8±1.7 | 48.5±1.2 | 48.8±1.2 | 49.3±1.5 | 51.2±0.8 | 49.3 |
| | | SHOT-best | 78.4±1.6 | 81.1±1.4 | 81.8±1.1 | 82.3±1.2 | 83.2±1.0 | 81.4 |
| | | SHOT-strong | 76.2±1.9 | 76.2±1.6 | 75.7±1.1 | 75.2±0.9 | 80.1±0.8 | 76.7 |
| | Multiple | SHOT-ens | 79.3±1.5 | 80.1±1.1 | 80.2±1.0 | 82.1±1.1 | 83.8±0.7 | 81.1 |
| | | TOHAN-ens | 93.1±1.2 | 96.5±0.9 | 97.1±0.7 | **97.5±1.2** | 97.7±1.0 | 96.4 |
| | | DECISION | 95.6±1.2 | 95.3±1.0 | 95.3±0.8 | 95.3±1.0 | 95.6±1.2 | 95.4 |
| | | Bi-ATEN | 95.2±1.9 | 96.1±1.7 | 96.5±0.9 | 97.1±1.1 | 97.7±1.3 | 96.5 |
| | | HiFE (ours) | **96.6±0.6** | **97.3±0.7** | **97.7±0.5** | **97.5±0.4** | **98.0±0.6** | **97.4** |
| $W \to D$ | Single | SHOT-worst | 60.9±1.2 | 57.8±1.1 | 55.9±1.1 | 55.5±0.9 | 53.9±0.8 | 56.8 |
| | | SHOT-best | 98.6±1.3 | 98.6±1.5 | 98.4±1.2 | **99.6±0.8** | 99.6±1.8 | 99.0 |
| | | SHOT-strong | 98.6±0.5 | 98.6±0.4 | 98.2±0.5 | 98.8±0.3 | 99.6±0.9 | 98.8 |
| | Multiple | SHOT-ens | 99.2±0.4 | 99.2±0.3 | 99.3±0.4 | 99.5±0.3 | 99.2±0.3 | 99.3 |
| | | TOHAN-ens | 99.2±0.2 | 99.4±0.3 | 99.3±0.2 | 99.4±0.4 | 99.5±0.2 | 99.4 |
| | | DECISION | **99.5±0.6** | 98.9±0.6 | 99.3±0.3 | 99.2±0.4 | 99.2±0.1 | 99.2 |
| | | Bi-ATEN | 99.4±0.6 | **99.6±0.4** | 99.4±0.6 | 99.3±0.3 | 99.6±0.4 | 99.5 |
| | | HiFE (ours) | **99.5±0.6** | **99.6±0.1** | **99.6±0.1** | **99.6±0.3** | **99.9±0.2** | **99.6** |

Table 9: Classification accuracy±standard deviation(%) on six adaptation tasks of office datasets. A, D, and W are abbreviations of Amazon, DSLR, and Webcam. The suffixes of -best and -worst refer to the best and worst results after adapting each single source hypothesis. The suffixes -strong and -ens refer to the result after adapting the strong hypothesis and the ensemble of all adapted hypotheses, respectively. Results of SHOT (Liang et al., 2020), TOHAN (Chi et al., 2021), DECISION (Ahmed et al., 2021) and our HiFE are presented. The highest accuracy is marked in bold.

| Tasks | Methods | Number of Target Data per Class | | | | | Average |
|---|---|---|---|---|---|---|---|
| | | 1 | 2 | 3 | 4 | 5 | |
| $D, W \rightarrow A$ | SHOT-ens | 52.8±1.2 | 53.4±1.3 | 55.7±2.1 | 59.1±1.8 | 60.9±1.2 | 56.4 |
| | TOHAN-ens | 57.1±1.4 | 61.7±1.2 | 63.2±1.1 | 64.1±0.9 | 65.1±0.8 | 62.2 |
| | DECISION | 53.3±1.7 | 54.1±2.2 | 55.2±2.1 | 55.9±1.6 | 57.1±1.2 | 55.1 |
| | Bi-ATEN | 55.4±1.3 | 55.3±2.1 | 58.7±1.6 | 60.1±1.7 | 62.5±1.5 | 58.4 |
| | HiFE (ours) | **63.2±0.5** | **65.5±0.6** | **67.8±0.6** | **68.1±0.7** | **68.3±0.4** | **66.6** |
| $D \rightarrow A$ | HiFE (ours) | 61.8±1.0 | 64.7±0.7 | 67.2±0.6 | 66.8±1.0 | 67.5±0.9 | 65.6 |
| $W \rightarrow A$ | HiFE (ours) | 62.5±2.5 | 65.1±1.5 | 64.4±1.2 | 64.3±0.9 | 64.8±0.8 | 64.2 |

Table 10: Classification accuracy±standard deviation (%) on **multi-domain multi-hypotheses** adaptation task from the office dataset DSLR (D), Webcam(A) to Amazon (A). SHOT-ens (Liang et al., 2020), TOHAN-ens (Chi et al., 2021) refers to the result of the ensemble of all corresponding adapted source hypotheses. The highest accuracy (%) is marked in bold. We also list the results of single-source adaptation tasks $D \rightarrow A$ and $W \rightarrow A$ of HiFE for further comparison.

## G    HiFE for Partial Few-Shot Hypothesis Adaptation

We extend HiFE to a partial few-shot hypothesis adaptation scenario. The office dataset contains three domains, including Amazon (A), DSLR (D), and Webcam (W), with each domain containing images of 31 object categories. We use images from the first 17 classes as the target domain and images from all 31 classes as the source domain. In this way, we obtain six partial domain adaptation tasks. HiFE is also better than the SOTA in the partial few-shot hypothesis adaptation scenario (see details in Table 11).

| Tasks | Methods | Number of Target Data per Class | | | | | Average |
|---|---|---|---|---|---|---|---|
| | | 1 | 2 | 3 | 4 | 5 | |
| $A \rightarrow D$ | SHOT-ens | 82.5±2.2 | 82.8±1.9 | 83.0±1.1 | 84.6±1.6 | 87.0±1.0 | 84.0 |
| | TOHAN-ens | 83.6±2.1 | 83.6±2.0 | **89.2±1.8** | **91.1±1.3** | 90.7±1.6 | 87.6 |
| | DECISION | 83.0±2.5 | 83.3±2.1 | 84.7±1.6 | 86.3±1.9 | 87.0±1.2 | 84.9 |
| | HiFE (ours) | **89.0±2.1** | **89.6±1.3** | 89.2±2.2 | 89.7±1.4 | **91.8±1.1** | **89.9** |
| $D \rightarrow A$ | SHOT-ens | 61.0±1.3 | 63.4±2.1 | 62.8±1.6 | 67.7±1.2 | 68.1±2.1 | 64.7 |
| | TOHAN-ens | 64.2±2.1 | 66.5±1.5 | 71.0±1.4 | 70.6±0.9 | 69.6±1.1 | 68.4 |
| | DECISION | 58.7±1.4 | 58.4±1.7 | 59.7±2.0 | 61.1±1.3 | 61.0±1.9 | 59.8 |
| | HiFE (ours) | **65.3±2.0** | **77.0±1.5** | **76.9±1.1** | **76.2±0.8** | **77.4±2.1** | **74.6** |
| $W \rightarrow A$ | SHOT-ens | 55.9±2.0 | 59±1.8 | 60.0±1.6 | 64.2±0.9 | 65±1.2 | 60.8 |
| | TOHAN-ens | 59.4±0.7 | 60.3±0.8 | 62.1±1.9 | 64.2±1.3 | 70.6±1.4 | 63.3 |
| | DECISION | 54.5±1.2 | 55.9±1.9 | 58.0±3.0 | 60.0±2.2 | 63.6±2.0 | 58.4 |
| | HiFE (ours) | **67.1±1.8** | **72.0±1.0** | **71.3±2.1** | **71.9±2.1** | **72.2±0.9** | **70.9** |

Table 11: Classification accuracies (%) on six partial few-shot hypotheses adaptation tasks of office datasets. A, D, and W are abbreviations of Amazon, DSLR, and Webcam. Results of SHOT (Liang et al., 2020), TOHAN (Chi et al., 2021), DECISION (Ahmed et al., 2021) and our HiFE are presented. The bold value represents the highest accuracy (%).

## H    Complexity Analysis

We compare the time complexity and parameter size of our framework HiFE, a single model, and the ensemble of all models using DECISION in Table 12. The experiments are conducted using the digit dataset for the SVHN to Mnist task with a LeNet model backbone. The batch size, the number of target samples per class, and the number of pre-trained models are set to 128, 7, and 8, respectively. HiFE, being a multi-hypotheses adaptation approach, incurs additional computational costs compared to the single model baseline. Specifically, the cost is approximately 8 times higher, corresponding to the number of pre-trained models. While HiFE has only slightly more parameters than the ensemble method DECISION, it is significantly more efficient in its usage. Overall, HiFE strikes a better balance between performance and efficiency for multi-hypotheses adaptation.

| Model | Acc (%) | Params (M) | Memory (M) | Speed (samples/s) |
|---|---|---|---|---|
| LeNet (Single) | $46.2 \sim 82.2$ | 2.1 | 16 | 142201 |
| DECISION (Multiple) | 83.5 | 16.1 | 128 | 17851 |
| HiFE(our) (Multiple) | 89.2 | 16.6 | 131 | 17758 |

Table 12: Performance and complexity of HiFE, ensemble method DECISION and the LeNet models.

