# OpenReview forum: "HiFE: Hierarchical Feature Ensemble Framework for Few-shot Hypotheses Adaptation"
_TMLR — Accepted by TMLR_

### Review · Reviewer_Qbkv · 2024-06-26

**Summary Of Contributions:**

The authors introduce a hierarchical feature ensemble framework which is useful for source-free domain adaptation.  This approach combines features from both strong and weak leaners which enables the use of incorporation of features learned from weak, intentionally underfit source models.  These models enable greater generalization for the target domain.  This framework shows enhanced performance on few-shot hypothesis adaptation scenarios.

**Audience:**

Yes

**Claims And Evidence:**

No

**Requested Changes:**

While overall the writing is relatively clear, some of the sentences are a bit awkward.  For example, page 1 opening sentence of 2nd paragraph "as indicated by the results of an experiment".  Is this referring to a specific experiment (if so cite), or is this referring to a subsequent abstract test the person performing the adaptation makes?  Sentences like this could be clearer.  The following sentence "we adapt models...of SVHN to the target task of MNIST" makes that sound as if it is the ultimate goal of the approach, which it is not.  The method may be demonstrated on this task ("we demonstrate our approach on the toy example of ...."), but just a change in the language would help frame this better.  The authors reference a meaningful example within the medical space, but it's not sufficiently explored or explained, IMO.

Similarly Figure 1 is discussed specifically in terms of the SVHN to MNIST transfer, so it's unclear whether the details are related to this specific task, or more broadly.  In general, I would suggest Figure 1 be broadly capture the approach so that the readers know immediately what the problem, solution, and impact is.  Focusing on such a narrow result leaves the reader with a lot of questions after the introduction.  Futhermore, Figure 1 is not particularly clear in illuminating what problem is being solver or how this particular problem solves it.  One alternative could be to pick a tangible relevant examples from these different domains and show how they are integrated.

Definition 2: "M is the hypothesis number".  "m" is the hypothesis number/ID and "M" is the number of hypotheses.

Table 1 is very unclear

To better speak to the impact in areas with privacy concerns which the authors raise, I think it is important to show these results on a real world or difficult dataset and not just toy datasets.

**Strengths And Weaknesses:**

STRENGTHS:
* Leveraging data from a variety of sources to improve transfer learning has obvious tangible benefits to the community
*Performance is evaluated on several different datasets across digits, office, and broad image classification.  Comparison to several baselines are made- these are reasonably chosen.
* Results are shown with uncertainty values
* Ablation studies capture the impact of the DeCL loss and number of hypotheses.

WEAKNESS:
* While the writing is clear grammatically, there could be improvements made to the overall flow and some of the sentences to improve understanding.  Specifics are given in the Requested Changes.
*The main contribution is around proposing multiple weak hypotheses as opposed to multiple strong or a single weak hypothesis.  I'm not sure I would consider this a novel "problem" as the authors present it.  Ultimately the goal is to perform source to target transfer.  Whether that is done with a single or multiple (weak or strong) hypothesis, doesn't make it a new task, but simply a different approach.
* The importance of the problem is introduced in the paper as having privacy importance for fields like medical.  However, none of the experiments address this application.  I would like to see this method tested on a "hard" or "real world" dataset, instead of just academic datasets.
*From the ablation studies it's not clear if the recommendation around the number of weak hypotheses is specific to these task/data or whether a different task/dataset/amount of data would impact this choice.  Similarly is the limit on the performance of the weak hypothesis relative to the overall (potential) strength of that model, or is that unique to this data?
* All of the experiments revolve around image classification.  Is this the only task where this is valid?  What about other image tasks such as segmentation?  What about non-image tasks?  I think addressing this limitation is key for it delivering significant value to the community.

---

> ### Author Response · Authors · 2024-07-29
>
> Dear Reviewer,
>
> Thank you very much for your detailed and instructive feedback on our manuscript. We greatly appreciate the time and effort you have invested in reviewing our work. Below are our responses to your comments and the corresponding revisions we have made:
>
> 1.	**Rewrite Sentences to Improve Understanding**. We have revised the sentences you mentioned. We also polish the whole manuscript carefully to modify some sentences to improve clarity. E.g.:
>
>     a) This indicates that some weak hypotheses, although suboptimal for the SD, may possess underlying knowledge that could be instrumental in the TD. => This indicates that some weak hypotheses, although suboptimal for the SD, may contain underlying knowledge that could be beneficial for the TD.
>
>     b) In this experiment, we adapt models with different accuracies from the digit dataset SVHN to the target task MNIST. => This overfitting issue has been demonstrated by a pilot experiment in which models with varying accuracies from the digit dataset SVHN were adapted to the target task Mnist using a straightforward fine-tuning approach.
>
> 2.	**Experiments Revolve Around Image Classification**. For the current research, we focus on classification tasks. We acknowledge this as a limitation. Classification tasks are relatively widely-used and and straightforward, allowing researchers to study the effects of few-shot learning techniques more conveniently. While our current experiments are centered on classification tasks, we recognize the importance of extending few-shot adaptation methods to more complex tasks such as segmentation. We are actively exploring this potential in future work.
>
> 3.	**Table 1 Clarification**. Table 1 shows an example where we generate 12 source hypotheses under different validation accuracy ranges. We have added detailed captions and explanations to enhance understanding (Page 9).
>
> 4.	**More Difficult Dataset**. To make the result more convincible , we have added experiments on the VisDA-C dataset. VisDA-C is a demanding large-scale benchmark designed primarily for synthesis-to-real object recognition tasks. The source domain comprises 152,000 synthetic images created by rendering 3D models, whereas the target domain includes 72,000 real object images sourced from Microsoft COCO. In this experiment, compared with previous methods, HiFE excels, particularly when fewer samples are available. For example, when the number of target samples per class is 10, HIFE （76.1\%）outperforms the SOTA methods Bi-ATEN and TOHAN (72.5\%) by 3.6\%. This finding indicates that HiFE effectively leverages weak hypotheses to enhance adaptation accuracy under conditions of data scarcity. The detailed results and analysis for this dataset have been added on Page 9.
>
> 5.	**Definition 2 Clarification**. We have corrected the notation in Definition 2 to clearly state that "m" is the hypothesis ID and "M" is the number of hypotheses.
>
> We have updated the manuscript. The main changes have been highlighted in blue for your convenience. Once again, thank you for your insightful comments and constructive criticism. We believe these revisions address your concerns and significantly improve the clarity and impact of our manuscript. We look forward to your further comments and hope that our revised manuscript meets your expectations.
>
> Best regards,
>
> The Authors

---

### Review · Reviewer_s9f7 · 2024-07-05

**Summary Of Contributions:**

1. Points out that in the few-shot hypothesis adaptation task, using only one single strong model will lead to model overfit SD and poor performance in TD.
2. Proposed a new problem: FHAW to enhance the diversity of source models.
3. Proposed a new framework to solve FHAW problem and achieve excellent performance on multiple benchmarks.

**Audience:**

Yes

**Broader Impact Concerns:**

Nothing.

**Claims And Evidence:**

Yes

**Requested Changes:**

1. The feature that needs to be merged needs more discussion. Why select highest similarity rather than similarity more than a threshold?

**Strengths And Weaknesses:**

Strengths:
1. The author has made sufficient analysis and deductions of the theory in the paper.
2. The author has conducted sufficient ablation study on proposed framework to prove method efficiency.
3. The paper has cited enough relevant paper to prove the reliability of the viewpoint.
4. The author provides the complete code for verification
5. The writing of whole paper is excellent.

Weaknesses:
Some details need more explanation. e.g. 'two or more features with the highest similarity and merge them into a new feature for the next layer'

---

> ### Author Response · Authors · 2024-07-29
>
> Dear Reviewer,
>
> Thank you very much for your instructive feedback on our manuscript. We greatly appreciate the time and effort you have invested in reviewing our work.
>
> Regarding your question about the feature merging criteria, we would like to clarify the following: each time, we select the features with the highest similarity to merge. We do not set a similarity threshold because our approach aims to engage all input features, merging them layer by layer until only one feature remains. This ensures that no features are left unmerged and that the merging process is comprehensive. We have also modified the manuscript on pages 6-7 to provide a clearer explanation of this process. We hope this revision addresses your concerns and makes our methodology more understandable. The main changes have been highlighted in blue in the manuscript for your convenience.
>
> Once again, thank you for your insightful comments and constructive criticism. We look forward to your further comments and hope that our revised manuscript meets your expectations.
>
> Best regards,
>
> The Authors

---

### Review · Reviewer_1C1M · 2024-07-10

**Summary Of Contributions:**

The paper is about domain adaptation: authors formulate a new problem on how to use set of weak models (hypothesis) in addition to strong model trained on one source domain to perform finetuning (domain adaptation) on the target domain with limited labeled data. Authors advocate that this formulation of the problem is realistic and practical - we can store several snapshots of the same model and providers can give them w/o leaking the source private data.

Authors show that 1) usage of weak models is helpful and ensembling of them and strong model improves domain adaptation to the target domain with limited labeled data 2) authors propose a new block how to ensemble the weak and strong models via residual hierarchical ensembling in the feature space. Evaluation is purely empirical with set of the small datasets like MNIST, SVHN, CIFAR-10, STL-10 and USPS and Resnet / LetNet models showing that proposed framework outperforms other SOTA models.

**Audience:**

Yes

**Broader Impact Concerns:**

I don't see ethical concerns as the paper about general ML method which targets improvements for the generalization on out of domain data.

**Claims And Evidence:**

No

**Requested Changes:**

The paper itself is well written and reads smoothly and clearly. I have several suggestions for couple of the places for the readability and formatting:
- In the introduction and contribution it is worth maybe mention ensembling in general and compare with it, otherwise while reading I am in question what you did exactly as this is known in the boosting / ensembling that weak models together improve generalization.
- page 3 "to encoder the source knowledge" -> "to encode ..."
- page 3 "amounts of unlabeled" -> "amount of ..."
- MNIST is used in many places as "mnist" or "Mnist"
- page 4 "In FHA, the research ..." there is missing verb - so sentence in unclear from English language.
- notation $h_\theta$ and $\theta_h$ - a bit misleading as the use is mixed. Maybe use only  $\theta_h$?
- "Which feature to merge" - is the merging done only between features from the previous layers? I think a bit clear description is needed here. Do we also recompute grouping after every layer?
- Is merging order depends on the input itself? I mean that for every sample the merging order will be different? It means it is not parallelizable at all, which makes this **really impractical** as it is not batched then. Any comments on this?
- One point authors need to highlight is that their method definitely reduce std
- "matrixes" -> "matrices"
- Appendix "we shown" -> "we showed"
- Appendix theorems and lemmas - not clear if this is contribution from authors or it is prior work
- "In our HiFE, We" -> "In our HiFE, we"
- What is speed for the framework? - mainly for the inference - what is additional overhead?

Regarding the empirical analysis I have the following thoughts:
- Fig 1 motivation plot - I wanna to see the y-axis to be exact accuracy numbers with std, as otherwise this could be misleading and maybe models are different in a very little performance in ranking within std even. Also this plot I found controversial as you are showing that model 2/4 are top, while model 1/3 are not and there is no really correlation between model performance on the source domain and the domain adaptation performance
- Definitions of weak and strong models is strange as it is based only on the training loss and not validation loss. During the empirical analysis authors in contrary used the accuracy range (I presume computed on the validation) which then contradicts the given definition. I think some work here should be done on clear understanding what weak models should be taken and maybe it should be level of overfitting (diff between train and test loss) and also the validation accuracy itself. In the main text authors speak a lot about overfitting, but this metric never used / analysed in the empirical part of the paper.
- As there is no new theoretical proofs / results in the paper and it is purely empirical (which is totally fine) I believe empirical analysis should be done with more realistic data too, not only small outdated datasets like MNIST and CIFAR and small old models like Lenet. I agree that a lot of classic models are used in many applications like MRI where the data are very limited and we don't need to have fancy models. However, either you consider more practical datasets and close to current models usage in the applications or switch to e.g. iWildCam setting or at least Imagenet + CIFAR. Also very simple setting is considered like 10 classes only.
- There is no understanding what happens in more realistic scenario of more classes and moreover different distribution of classes in the source domain and target domain (e.g. heavily unbalanced classes in the source domain).
- There is no ablation why only strong encoder is updated, and why all weak are kept?
- What is performance if we use simple ensemble of models w/o finetuning on the target domain?
- Deeper analysis / results discussion on how we depend on the number of samples in the target domain - it seems e.g. it doesn't matter what weak models to take if there are few samples, while for more samples it is better to have stronger weak models
- There is no discussion on how to select the weak classifiers at all, and from Tables it seems we depend on it a lot.

**Strengths And Weaknesses:**

**Strengths**
- idea on exploiting the weak models along with strong for domain adaptation
- reasonable problem formulation for domain adaptation
- new block on how to combine features from different models

**Weaknesses**
- empirical analysis is weak (see requested changes) as model are very outdated and datasets are very simple (having that there is no theoretical analysis)
- definitions on the weak and strong are unrelated to the empirical analysis and how weak models are defined. There is no connection to overfitting, which make it questionable how to select the weak models (see requested changes)
- no analysis / ablation how to select the weak models (see requested changes)
- the method seems not to be batched for the inference and input dependent (for the merge operation) - so in practice it will be slow for the data as e.g. images (see requested changes)

---

> ### Author Response · Authors · 2024-07-29
>
> Dear Reviewer,
>
> Thank you very much for your detailed and instructive feedback on our manuscript. We greatly appreciate the time and effort you have invested in reviewing our work. We have carefully considered all your recommendations and have made the necessary revisions to address each of your points.
>
> 1.	**Mention Ensembling in Introduction**. We have incorporated a discussion on ensembling in the introduction (Page 1-2). Motivated by your opinion, we have also updated our motivation figure  (Figure 1). We agree that adding this part makes our motivation clearer and helps to contextualize our work within the broader framework of boosting and ensembling techniques.
>
> 2.	**Notation**. Regarding your concern about the notation h_\theta  and  \theta_h, we understand that the notation might be a bit confusing. Therefore, we have decided to remove  h_\theta and only keep \theta_h to distinguish the parameters of different hypotheses. We believe this change will enhance clarity and avoid any potential misunderstandings (Page 4).
>
> 3.	**Merging Order and Input Dependency**. The merging is done only between features from the previous layer. We determine which features to merge based on the input base on the first batch of samples. Once the merging network is established, it remains fixed for further stable fine-tuning on target samples. This approach ensures that while the initial computation of the merging network is input-dependent, the subsequent processing is consistent and parallelizable across different samples. We revised the manuscript to provide a clearer description of this process (Page 6-7).
>
> 4.	**Reduction of Standard Deviation**. We have added a discussion on page 9 to emphasize our method effectively reduces standard deviation, thereby indicating a more consistent and reliable performance of our HiFE.
>
> 5.	**Theorems and Lemmas in the Appendix**. We would like to clarify that these theorems are from a previous ork. We included these theorems because they are related to our research and provide theoretical support for our approach. Specifically, they demonstrate that adding weak hypotheses increases the diversity of the source models and offers opportunities to reduce the error bound.
>
> 6.	**Speed of the Framework**. Since the merging network is simple and not time-consuming, the speed primarily depends on the number of source encoders. The final computational cost is approximately n times the original single source model's computational cost, where n is the number of source hypotheses. Although this computational cost is higher than that of the original single model, we believe that in situations where there is a shortage of data and no other approaches are available to improve target accuracy, our method provides a viable alternative to enhance adaptation performance.
>
> 7.	**Definitions of Weak and Strong Models**. We would like to clarify that the definition is actually based on the validation accuracy. We divide the source data into training and validation sets, and the selection of weak and strong hypotheses is based on the accuracy on the validation set from the source domain. Additionally, we discuss the overfitting situation in the Sec 5.2 (Page 9). This can be realized by comparing the results of SHOT-strong and SHOT-best. Typically, the accuracy of SHOT-strong is worse than that of SHOT-best, indicating that the strong source model overfits the source domain. In contrast, some weak models may achieve better results on the target domain, highlighting the overfitting issue.
> 8.	**Use of More Realistic Datasets and Models**. We have made the following updates to our experiment analysis:
>
>         a) For the dataset, we have added the  VisDA-C, a demanding large-scale benchmark designed primarily for synthesis-to-real object recognition tasks. The source domain comprises 152,000 synthetic images created by rendering 3D models, whereas the target domain includes 72,000 real object images sourced from Microsoft COCO. In this experiment, compared with previous methods, HiFE excels, particularly when fewer samples are available. For example, when the number of target samples per class is 10, HIFE （76.1\%）outperforms the SOTA methods Bi-ATEN and TOHAN (72.5\%) by 3.6\%. This finding indicates that HiFE effectively leverages weak hypotheses to enhance adaptation accuracy under conditions of data scarcity. The detailed results and analysis for this dataset have been added on Page 9.
>
>         b) For the models, in addition to LeNet, we have adopted ResNet-18, ResNet-50, and ResNet-101 as the backbones for the CIFAR-10/STL-10, Office, and VisDA-C datasets, respectively. This ensures that our analysis includes more widely-used backbones.
>
> 9.	**Simple Ensemble of Models W/o Fine-Tuning**. Since our approach contains weak hypotheses with relatively low source accuracy, traditional ensemble methods without fine-tuning are likely to result in very low performance.
>
> (see the next post)

---

> ### Author Response · Authors · 2024-07-29
>
> (continuous)
>
> 10.	**Analysis on the Number of Samples in the Target Domain**. We have conducted additional analysis on the VisDA-C experiment. Our new analysis shows that as the number of target samples increases to 50 per class, the performance gap narrows, suggesting that the dependency on the source model diminishes with more target samples given. A detailed comparison of HiFE with previous methods reveals that HiFE excels when fewer samples are available. Specifically, under the situation where only a few target samples are given, the adaptation accuracy depends more on the source models, and the weak source models provide more help (Page 9).
>
> 11.	**Selection of Weak Classifiers**: We have discussed this topic in detail in our manuscript. Specifically, the selection of the number of weak classifiers is addressed in "Ablation study on the number of weak hypotheses." Additionally, the selection of the appropriate accuracy range for weak classifiers is discussed in "Ablation study on the accuracy range of weak hypotheses" (Page 10-11).
>
> We have updated the manuscript. The main changes have been highlighted in blue for your convenience. We believe that these changes have significantly strengthened our manuscript, and we are grateful for your guidance in this process. Once again, thank you for your thorough review and constructive criticism. We look forward to your further comments and hope that our revised manuscript meets your expectations.
>
> Best regards,
>
> The Authors

---

> > ### Comment · Reviewer_1C1M · 2024-07-30
> > **Thanks for the clarifications!**
> >
> > Dear Authors,
> >
> > Thanks for the updated manuscript, additional experiments and clarifications. Please find below my comments on some of your responses:
> >
> > > Merging Order and Input Dependency. The merging is done only between features from the previous layer. We determine which features to merge based on the input base on the first batch of samples. Once the merging network is established, it remains fixed for further stable fine-tuning on target samples. This approach ensures that while the initial computation of the merging network is input-dependent, the subsequent processing is consistent and parallelizable across different samples. We revised the manuscript to provide a clearer description of this process (Page 6-7).
> >
> > Ahh, this makes sense and then resolve my main concern about non-parallel computations. But what batch size do you use? How do we depend on this first batch? Assume I try two different random batches to define the merging procedure - so how are we stable with respect to it?
> >
> > > Theorems and Lemmas in the Appendix.
> >
> > This should be clearly stated in text to remove confusion as I had.
> >
> > > Speed of the Framework.
> >
> > I am ok if this is n times slowly and agree this depends on the application, but this should be provided as analysis in the paper, so that reader knows the tradeoffs.
> >
> > > Definitions of Weak and Strong Models.
> >
> > Then the definition should be fixed in the beginning as it is not aligned with it (Def 2 and Def 3) where loss is used and seems on the training data (not validation).
> >
> > > Use of More Realistic Datasets and Models.
> >
> > Thanks for adding this!
> >
> > > Simple Ensemble of Models W/o Fine-Tuning.
> >
> > Agree - but can you run this ablation to have solid empirical confirmation? Because Random Forest works well in many-many applications, so I could argue that many weak models could have similar effect (though random forest has bagging). Also why not finetune all weak model too then but only strong? This choice is not obvious to me.
> >
> > > Analysis on the Number of Samples in the Target Domain.
> >
> > Thanks for the experiments, and results make sense as well as nice to see now the conditions where your method is strong.
> >
> > > The main changes have been highlighted in blue for your convenience.
> >
> > Thank you very much! Will have a look at the updated manuscript in the upcoming days and will let you know any further comments if any.
> >
> > Reviewer.

---

> > > ### Comment · Reviewer_1C1M · 2024-08-02
> > > **Comments on the revision**
> > >
> > > Dear Authors,
> > >
> > > I had a look at the revised text (sorry if missed anything, but mainly checked the blue colored text). Please find below my thoughts (and just in case the provided responses and additional experiments almost resolved my concerns).
> > >
> > > > Definitions of weak and strong models is strange as it is based only on the training loss and not validation loss. During the empirical analysis authors in contrary used the accuracy range (I presume computed on the validation) which then contradicts the given definition. I think some work here should be done on clear understanding what weak models should be taken and maybe it should be level of overfitting (diff between train and test loss) and also the validation accuracy itself. In the main text authors speak a lot about overfitting, but this metric never used / analysed in the empirical part of the paper.
> > >
> > > I still see that in definitions there is the point about empirical risk, but this doesn’t imply the validation performance, I still read it as the train one (maybe I am wrong with respect to the classic formulation/usage of empirical risk minimization? - maybe instead of definitions just explicitly say about selecting the weak models based on the validation accuracy in the method description? this is more helpful at least for me when I read. If definitions are needed for theory part in Appendix, then move to it?
> > >
> > > > There is no ablation why only strong encoder is updated, and why all weak are kept?
> > >
> > > Any thoughts / comments on this?
> > >
> > > > Simple Ensemble of Models W/o Fine-Tuning. Since our approach contains weak hypotheses with relatively low source accuracy, traditional ensemble methods without fine-tuning are likely to result in very low performance.
> > >
> > > I still think this could be possible =) As from Table 11 you show that weak models are needed, and you don’t train them, only strong model - so I could expect that this doesn’t hurt your procedure, so maybe it is going to work? But otherwise it is interesting baseline for your idea, explaining e.g. that we need to finetune some models, and the choice why only strong model (e.g. reducing amount of resources, as only one model is finetuned and it is enough).
> > >
> > > If you still think that simple ensemble will not work due to weak models - then why do you finetune strong model and not all weak models instead? It should give stronger results probably (giving apart computational issue of finetuning all of them).
> > >
> > >  > Theorems and Lemmas in the Appendix. We would like to clarify that these theorems are from a previous ork. We included these theorems because they are related to our research and provide theoretical support for our approach. Specifically, they demonstrate that adding weak hypotheses increases the diversity of the source models and offers opportunities to reduce the error bound.
> > >
> > > Looking at the revision, I still think that it is good to point that this is just for the overview and readability. Up to you - minor for me.
> > >
> > > > Selection of Weak Classifiers: We have discussed this topic in detail in our manuscript. Specifically, the selection of the number of weak classifiers is addressed in "Ablation study on the number of weak hypotheses." Additionally, the selection of the appropriate accuracy range for weak classifiers is discussed in "Ablation study on the accuracy range of weak hypotheses" (Page 10-11).
> > >
> > > Could you point on the heuristic you have in mind given the Tables? My main question was (sorry for confusion) is how based on that I can select what weak models I need to take? it seems it is very dependent on the data / validation accuracy. If I have new problem how would you summarize recommendation then? (if we would robust - no question, but here the performance is very dependent in Table 7).
> > >
> > > > Regarding the merging procedure
> > >
> > > Could you point how it is stable now with respect to this first batch? What happens if another batch is used and how do we depend on the batch size here? e.g. what minimal number of samples I need to have in practice?
> > >
> > > Thanks for the revision and discussion,
> > >
> > > Reviewer.

---

> > > > ### Comment · Reviewer_1C1M · 2024-08-08
> > > > **Any more comments?**
> > > >
> > > > Dear Authors,
> > > >
> > > > Any more comments on my extra questions?
> > > >
> > > > Thanks!

---

> ### Author Response · Authors · 2024-08-08
>
> Dear Reviewer:
>
> We apologize for the delay in responding to your additional questions and thank you for your patience. We greatly appreciate your valuable feedback and insightful questions. Below, we address each of the points you raised:
>
> **1.	Merging order and input dependency.** In our setting, we focus on few-shot learning. When training with our approach, HiFE, the batch size is often larger than the total sample size. For instance, in a digit dataset where each class has 7 target samples and there are 10 classes, the maximum sample size is 70. This is smaller than our batch size of 128 when using LeNet, allowing us to use a very large batch size. However, when the sample size exceeds the batch size, such as when using ResNet101 where we reduce the batch size to 32, we address this by repeating each experiment five times. In each experiment, the first batch of images is chosen randomly. By averaging the results across multiple runs with different random batches, we ensure that our findings are not overly influenced by the initial batch selection.
>
> Your question has prompted us to consider a more practical approach. In realistic scenarios, it is indeed beneficial to use all available input images when constructing the model. Given the few-shot setting, the total number of images is manageable. Therefore, in future work we propose passing all images through the network initially to construct the model.
> Thank you once again for your valuable feedback. It has helped us refine our approach and consider more practical implementations.
>
> **2.	Theorems and Lemmas in the Appendix.** I have reviewed the Theorems section and ensured that each mentioned theorem is properly cited. The changes have been highlighted in orange for your convenience.
>
> **3.	Speed of the framework.** We have added a section in the Appendix to discuss the complexity of our framework. Please refer to Section H, "Complexity Analysis," for a detailed examination of the computational trade-offs involved.
>
> **4.	Only fine-tuning the strong encoder, why not fine-tune all weak models too?** We have added an ablation study on Page 12 (highlighted in orange). We conducted several experiments with different updated source encoders. The results show that updating most of the weak encoders decreases the final performance. The possible reason is that few-shot target samples are insufficient to guide the fine-tuning of both weak and strong encoders together.
>
> Although there may exist a weak hypothesis that is better suited for the target domain, it is unrealistic to identify this hypothesis beforehand without testing sets. Therefore, the most suitable approach is to only update the final strong hypothesis while keeping all other weak hypotheses unchanged among the source encoders. By designing and updating the hierarchical feature ensemble module to balance the merging weights among all the weak and strong encoders, we can generate the optimal final model.
>
> **5.	Selection of weak classifiers.** In the "Ablation study on the number of weak hypotheses," we observed that incorporating multiple weak hypotheses generally enhances performance. Additionally, in the "Ablation Study on the accuracy range of weak hypotheses," we found that as the accuracy range of the weak hypotheses increases, the performance after adaptation improves. However, it is important to note that weak models with an accuracy lower than 55% should not be included, as they may negatively impact the final performance.
>
> To summarize, our approach, HiFE, performs well when incorporating a larger number of weak hypotheses, provided that these hypotheses are not of very low accuracy. In our experiments, we selected 7 weak hypotheses as our base setting to balance performance and computational cost.
>
> In practical applications, the selection of weak hypotheses can be guided by the available computational resources and the performance of each weak hypothesis. We recommend choosing as many weak hypotheses as possible, ensuring that none have a lower source accuracy than a certain threshold. It is important to note that this threshold may vary depending on the specific application. We hope this clarifies our heuristic for selecting weak models and provides a robust guideline for applying HiFE to new problems.
>
> **6.	Definitions of weak and strong models.** Sorry for the confusion. We have revised the definitions of weak and strong hypotheses on Page 4, emphasizing that these definitions are based on the validation data.
>
> (see the next post)

---

> ### Author Response · Authors · 2024-08-08
>
> (continuous)
>
> **7.	Simple ensemble of models W/o fine-uning.** We did consider applying traditional ensemble approaches like random forests as one of our baselines. However, we found that traditional random forests are designed for decision trees and are well-suited for handling relatively simple tasks due to their inherent structure and the way they aggregate decisions from multiple trees. Our method, on the other hand, involves deep neural networks, which are fundamentally different in their architecture and learning capabilities. The model backbone we use cannot be easily replicated by the decision trees used in random forests.
> While random forests are powerful for certain types of problems, they are not inherently designed to leverage the sophisticated feature extraction and representation learning capabilities of deep neural networks.
>
> In our baselines, we have included several ensemble approaches that are more relevant to our method:
>
>         a) Naive ensemble approaches: These include SHOT-ens and TOHAN-ens, which involve sequentially adapting each hypothesis and then aggregating the prediction scores to identify the label with the highest value.
>
>         b) Typical ensemble methods in source hypotheses adaptation: We have included DECISION [1] and Bi-ATEN [2], which are commonly used in the source free domain adaptation area.
>
> By including these baselines, we aim to provide a more accurate and relevant comparison to demonstrate the effectiveness of our method.
>
> We hope that our responses have adequately addressed your concerns and provided clarity on the points you raised. Your feedback has been invaluable in helping us refine our approach and consider more practical implementations. We are grateful for your thorough review and constructive comments, which have significantly contributed to the improvement of our work. Thank you once again for your time and effort in reviewing our paper.
>
> Best regards
>
> The authors
>
> [1] Ahmed S M, Raychaudhuri D S, Paul S, et al. Unsupervised multi-source domain adaptation without access to source data[C]//Proceedings of the IEEE/CVF conference on computer vision and pattern recognition. 2021
>
> [2] Li X, Li J, Li F, et al. Agile Multi-Source-Free Domain Adaptation[C]//Proceedings of the AAAI Conference on Artificial Intelligence. 2024

---

> > ### Comment · Reviewer_1C1M · 2024-08-11
> > **Thanks for interesting ablations!**
> >
> > Dear Authors,
> >
> > Thanks a lot for the extra ablations, I found them really interesting! I think it is super useful finding on what exactly should be fine-tuned.
> >
> > Minor final suggestion on the writing only, all my concerns are resolved and I am happy with the final revision state!
> > - typo Table 7: UH for experiments 1-1 to 3-4 - should it be hypothesis h12? as it is the strong one and you always fine tune it?
> > - "Merging order and input dependency." - this is really strong argumentation from your side :) include this note about batch size and that your results are over several seeds - this is mainly for reproducibility and practical application when people will try to use it.
> >
> > Thanks again for the fruitful discussion, glad that you found discussion helpful!
> >
> > Reviewer.

---

> ### Author Response · Authors · 2024-08-11
>
> Dear Reviewer,
>
> Thank you very much for your positive feedback and for your constructive comments throughout the review process. We are pleased to hear that you found the additional ablations interesting and useful.
>
> We have corrected the typo in Table 7, changing "UH for experiments 1-1 to 3-4" to "hypothesis h12" as suggested. Thank you very much for pointing this out.
>
> Thank you once again for your valuable input and for the fruitful discussion.
>
> Best regards,
>
> The Authors

---

### Decision · Action_Editor_2rmJ · 2024-08-19

**Recommendation:** Accept as is

**Comment:**

This paper proposes HiFE which studies how to conduct effective domain adaptation by combining multiple weak checkpoints with a strong one. The reviewers generally find the paper to be well-written and the idea to be interesting and reasonable. The authors have done a good job answering the reviewers' questions and incorporating their very detailed feedback. The reviewers are generally satisfied with the paper after revision.

**Audience:**

Yes

**Claims And Evidence:**

Yes